# Hierarchical Entity-centric Reinforcement Learning with Factored Subgoal Diffusion

**Dan Haramati**[1]**, Carl Qi**[2]**, Tal Daniel**[3]**, Amy Zhang**[2]**, Aviv Tamar**[4†]**, George Konidaris**[1†]

[1]Brown University, [2]UT Austin, [3]Carnegie Mellon University, [4]Technion, † Equal advising
`dan_haramati@brown.edu;carlq@utexas.edu`

## Abstract

We propose a hierarchical entity-centric framework for offline Goal-Conditioned Reinforcement Learning (GCRL) that combines subgoal decomposition with factored structure to solve long-horizon tasks in domains with multiple entities. Achieving long-horizon goals in complex environments remains a core challenge in Reinforcement Learning (RL). Domains with multiple entities are particularly difficult due to their combinatorial complexity. GCRL facilitates generalization across goals and the use of subgoal structure, but struggles with high-dimensional observations and combinatorial state-spaces, especially under sparse reward. We employ a two-level hierarchy composed of a value-based GCRL agent and a factored subgoal-generating conditional diffusion model. The RL agent and subgoal generator are trained independently and composed post hoc through selective subgoal generation based on the value function, making the approach modular and compatible with existing GCRL algorithms. We introduce new variations to benchmark tasks that highlight the challenges of multi-entity domains, and show that our method consistently boosts performance of the underlying RL agent on image-based long-horizon tasks with sparse rewards, achieving over 150% higher success rates on the hardest task in our suite and generalizing to increasing horizons and numbers of entities. Rollout videos are provided at: `https://sites.google.com/view/hecrl`

## 1 Introduction

Planning to achieve complex, long-horizon goals in unstructured environments is a defining challenge at the core of Reinforcement Learning (RL) and sequential decision-making research. Goal-Conditioned RL (GCRL) (Kaelbling, 1993) provides a useful framework for training goal-reaching agents by facilitating generalization and knowledge transfer between goals and has proven promising when coupled with deep RL algorithms (Schaul et al., 2015). However, its effectiveness remains limited when faced with complex goal distributions and high-dimensional observation spaces, especially under sparse rewards (Park et al., 2025a).

Exploiting the subgoal structure in GCRL (i.e., the fact that intermediate states in a trajectory can also be treated as goals) has lead to significant progress in handling long-horizon sparse reward tasks via goal relabeling (Andrychowicz et al., 2017) and hierarchical learning (Levy et al., 2019; Chane-Sane et al., 2021; Bagaria et al., 2021). Park et al. (2023) proposed a two-level policy hierarchy extracted from a single goal-conditioned value function to address compounding approximation errors when learning with temporal difference objectives. The resulting algorithm, HIQL, offers a simple instantiation of hierarchy in the offline GCRL setting. However, as the recent OGBench benchmark (Park et al., 2025a) reveals, offline GCRL methods (including HIQL) continue to struggle with combinatorial state-spaces and image observations.

If we are to make progress towards applying GCRL to real-world problems and in domains such as robotics, our algorithms must scale to complex goal distributions and rich, high-dimensional observations such as images. Particularly challenging are scenarios where the environment contains multiple entities, where the state-space grows combinatorially with the number of entities. Examples for such domains include robotic object manipulation (Haramati et al., 2024), multi-robot path planning (Shaoul et al., 2025), autonomous driving (Vinitsky et al., 2018) and video games (Zambaldi et al., 2019; Delfosse et al., 2024). In this case, the underlying factored structure can be leveraged and incorporated as inductive bias to significantly simplify learning.

In this work, we seek to combine the benefits of subgoal hierarchy with factored structure to solve long-horizon tasks in multi-entity domains. A natural subgoal decomposition in some cases corresponds to modifying a subset of the entities, but as we show, state-of-the-art methods such as HIQL do not encourage this kind of sparsity. We propose a hierarchical entity-centric framework for offline GCRL that scales to image observations without requiring supervision. It leverages unsupervised object-centric representations and the factored structure in multi-entity environments to produce entity-factored subgoals, i.e., subgoals with sparse changes to entities compared to the current state.

We employ a two-level hierarchy where the low-level is a value-based GCRL agent and the high-level is a subgoal-generating conditional diffusion model. We show that the bias induced by entity-centric diffusion encourages entity-factored subgoals, which simplifies the subtask for the RL policy. Our method is designed to be modular and adaptive: the two levels are only related by the datasets they are trained on and are combined post-training via a subgoal generation procedure, which involves selective subgoal generation based on the value function. This allows simple integration with potentially any value-based GCRL algorithm.

We demonstrate agents that can effectively achieve long-horizon goals in combinatorially complex and high-dimensional observation spaces whilst learning from sparse rewards. We test our approach using novel variations of existing benchmarks to highlight the above challenges. Our method consistently boosts the performance of flat entity-centric GCRL agents, achieving more than a $150\%$ success increase on the most difficult task in our suite, and generalizes to increasing horizons and number of entities.

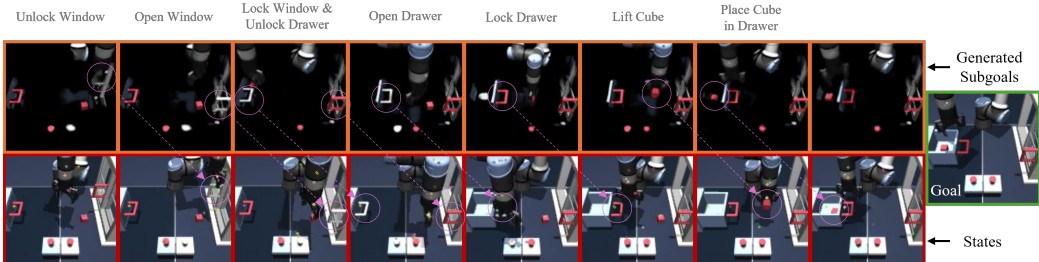

Figure 1: `Scene` **image-based factored subgoals**. *Bottom row*: environment image observations at the time of subgoal generation. *Top row*: Reconstructions of the generated latent factored subgoals. *Rightmost image*: goal image observation. *Circles and arrows*: circles on the top row highlight factors (excluding the arm) that are modified by the subgoals compared to the current observation in the image below them. Arrows leading to circles on the bottom row highlight the factors that are manipulated by the low-level RL policy, showing that the subgoals were achieved. Subgoal reconstructions only include the foreground of the scene captured by the Deep Latent Particles (DLP, Daniel & Tamar (2022)) representation. See App. for details on the task B and the DLP model A.3.

## 2 BACKGROUND & RELATED WORK

**Goal-conditioned Reinforcement Learning**: GCRL (Kaelbling, 1993) considers a Markov Decision Process $\mathcal{M} = (\mathcal{S}, \mathcal{A}, \mu, p, r)$, where $\mathcal{S}$ denotes the state space, $\mathcal{A}$ the action space, $\mu : \mathcal{P}(\mathcal{S})$ the initial state distribution, $p : \mathcal{S} \times \mathcal{A} \to \mathcal{P}(\mathcal{S})$ the environment transition dynamics, $r : \mathcal{S} \times \mathcal{G} \to \mathbb{R}$ the reward function and $\mathcal{G}$ the goal space. The objective is to learn a policy $\pi^* : \mathcal{S} \times \mathcal{G} \to \mathcal{A}$ that maximizes the expected discounted return $\mathbb{E}_\pi[\sum_{t=0}^\infty \gamma^t r_t]$ for a given goal distribution, $\gamma$ denoting the discount factor. Our work focuses on the offline setting, where the agent learns from a fixed dataset of suboptimal state-action trajectories. We assume that $\mathcal{G} = \mathcal{S}$ and a sparse reward $r_t(s_t, g) = \mathbf{1}\{s = g\} - 1$, i.e., 0 when at the goal and $-1$ otherwise.

**Hierarchical Goal-conditioned Reinforcement Learning**: Hierarchical RL enables reasoning over multiple timescales and levels of abstraction by exploiting temporal structure within sequential data (Klissarov et al., 2025). In the context of GCRL, many algorithms tackle long-horizon goal-reaching with various instantiations of subgoal hierarchies (Schmidhuber, 1991; Dayan & Hinton, 1992; Kulkarni et al., 2016; Vezhnevets et al., 2017; Nachum et al., 2018; Levy et al., 2019; Chane-Sane et al., 2021; Bagaria et al., 2021). Closely related to our work is HIQL (Park et al., 2023)

which extracts a two-level policy hierarchy from a single value function learned from offline data. In contrast to HIQL, we model the high-level policy using a diffusion model which as we show, is crucial for producing high-quality subgoals that correspond to valid states. We additionally incorporate entity-centric structure across the hierarchy to simplify learning in domains with combinatorial multi-entity state spaces and facilitate factored subgoals.

**Diffusion for Sequential Decision-making**: Diffusion models (Sohl-Dickstein et al., 2015; Ho et al., 2020) learn to generate data by reversing a process that gradually adds noise to clean samples, training a neural network to perform step by step denoising. The network is optimized by predicting the noise injected at each step and generates data by gradually denoising a sample from an initial noise distribution. These have become widely popular for their ability to capture complex multi-modal data distributions. Work applying diffusion models to decision-making can be coarsely divided to diffusion policies (Chi et al., 2023; Hansen-Estruch et al., 2023) which model distributions of actions (or action "chunks") and diffusion planners (also referred to as diffusers) (Janner et al., 2022; Ajay et al., 2023; Lu et al., 2025) which model distributions of state or state-action trajectories. Li et al. (2023); Chen et al. (2024) propose hierarchical diffusers that generate subgoal trajectories to condition lower-level diffusers for long-horizon tasks. Compared to the above, our method employs a high-level diffusion model without guidance to generate a single immediate subgoal for a GCRL policy and uses its value function for test-time subgoal selection. We adapt the entity-centric diffuser proposed in Qi et al. (2025), which was designed for goal-conditioned behavioral cloning, to generate entity-factored subgoals.

**Entity-centric Reinforcement Learning**: Entity-centric RL considers environments which can be described as a collection of entities i.e., $\mathcal{S} = \mathcal{S}_1 \otimes \mathcal{S}_2 \otimes ... \otimes \mathcal{S}_N$. This factored structure can simplify learning when incorporated in state representation and agent architecture (Sanchez-Gonzalez et al., 2018; Zadaianchuk et al., 2022; Sancaktar et al., 2022; Zhou et al., 2022) and facilitate compositional generalization (Lin et al., 2023). Obtaining factored representations of images that closely approximate the true state of the environment without supervision is not trivial and has been a subject of previous work. These either learn a representation concurrently with the decision-making modules (Zambaldi et al., 2019; Watters et al., 2019; Veerapaneni et al., 2020) or pretrain unsupervised object-centric representations (Lin et al., 2020; Locatello et al., 2020; Daniel & Tamar, 2022) for downstream RL (Zadaianchuk et al., 2021; Yoon et al., 2023; Haramati et al., 2024). We present a modular hierarchical framework that builds on Haramati et al. (2024) and enables long-horizon goal-reaching from sparse reward via factored subgoal diffusion.

See App. A for extended background and related work.

## 3 HIERARCHICAL ENTITY-CENTRIC REINFORCEMENT LEARNING

Our framework, Hierarchical Entity-Centric RL (HECRL), addresses two main aspects that hinder accurate value learning in multi-entity domains: reward propagation over long horizons 3.2 and combinatorial state complexity 3.3, both of which increase with the number of entities. We employ an entity-centric approach (Haramati et al., 2024; Qi et al., 2025) and propose a two-level hierarchy composed of a value-based GCRL agent and a *factored* subgoal-generating diffusion model. Our approach is modular and adaptive, making it compatible with various value-based GCRL algorithms. Additionally, it can handle image-based domains without access to the underlying factored environment state (e.g., robotic manipulation from pixels and video games) using unsupervised object-centric representations (e.g., DLPv2 (Daniel & Tamar, 2024), which we use in our experiments).

### 3.1 MOTIVATION: SIDESTEPPING COMPOUNDING VALUE APPROXIMATION ERROR

Leveraging factored structure in learned value functions improves performance in combinatorial state-spaces, but does not completely mitigate value approximation error. Temporal Difference (TD) learning (Sutton, 1988) leads to errors which compound over the timestep horizon. This error is more consequential in long-horizon sparse reward tasks where the reward must propagate through many steps of discounted TD updates. While the value function can still be useful for approximating distances between states and goals, it may not be as effective for goal-conditioned policy extraction: for states far away from a given goal, the approximation error can be larger than the fine difference in value between states that are one low-level action apart. Park et al. (2023) refer to this as the

value signal-to-noise ratio, which grows with the distance between states. This ratio defines an "effective radius" of states whose values provide a clear learning signal for policy extraction, which we will refer to as the *policy competence radius* and denote $R_\pi^V$ or $R$ for simplicity of notation. This quantity is not known a priori. Given an offline RL policy, our method aims to produce an immediate subgoal that is both *reachable* by that policy, that is, within $R_\pi^V$, and leads it closer to the goal. This process can be repeated until the goal is within the policy's reach. Two key aspects of our approach distinguish it from previous methods: (1) *Modularity and Test-time Flexibility*— we do not require any modifications to the underlying RL agent. We separately train a subgoal-generating conditional diffusion model and enforce a value-based reachability constraint at test-time (see Fig. 2, left, and App. D.2 for further discussion). (2) *Factored Subgoal Diffusion*—our entity-centric approach encourages subgoals with sparse modifications to state factors when the data supports it (see Fig. 2, right, and Sec. 4.5 for empirical results). Subgoals that require modifying few state factors are generally easier to reach when these factors (or subsets thereof) are independently controllable, making them favorable for conditioning the low-level policy.

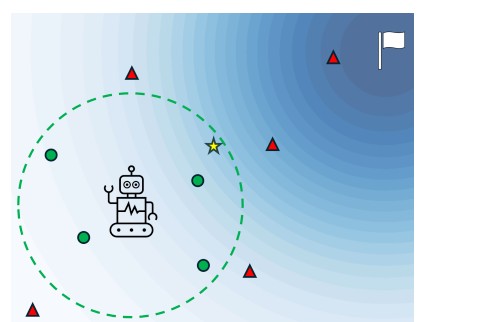 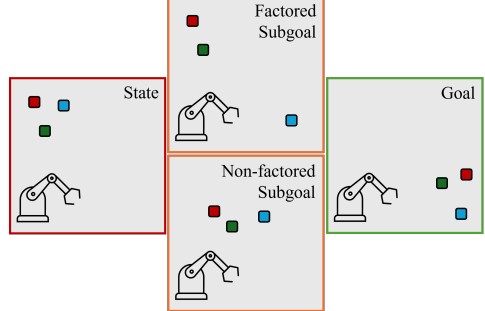

Figure 2: **Left: subgoal sampling illustration** (Alg. 1 lines 2–4). Robot - *agent*, Dashed circle - *competence radius*, Flag - *goal*, Red triangle - *discarded subgoal*, Green circle - *filtered subgoal*, Yellow star - *chosen subgoal*, Background color gradient - *value landscape*. **Right: factored subgoal illustration**. Factored subgoals make it easy to modify small subsets of entities which simplifies the subtask when factors are independently controllable. The subgoal images depict two states that are roughly the same distance from the initial state, where the top requires moving only the blue cube while the bottom requires moving all three.

## 3.2 THE SUBGOAL DIFFUSER

Our method assumes access to a trained value-based entity-centric GCRL agent consisting of a policy $\pi : \mathcal{S} \times \mathcal{G} \to \mathcal{A}$ and value function $V : \mathcal{S} \times \mathcal{G} \to \mathbb{R}$ as well as the offline dataset it was trained on. We remind the reader that in our setting $\mathcal{G} = \mathcal{S}$ but we maintain the $\mathcal{G}$ notation for readability.

We train a conditional diffusion-based subgoal generator $\mathcal{D} : \mathcal{S} \times \mathcal{G} \to \mathcal{G}$ on the offline RL dataset to fit the distribution of states that are at most $K$ timesteps away from a state $s$ given a goal $g$, where $g$ is a state sampled uniformly from future timesteps in the same trajectory as $s$. That is, given an offline state trajectory of length $T$: $(s_0, \ldots, s_T)$, we (1) uniformly sample a timestep $t$ from $[0, T-1]$ to select a state $s = s_t$; (2) uniformly sample a timestep $t_g$ from $[t+1, T]$ to select a goal state $g = s_{t_g}$; (3) set the training subgoal to $\tilde{g} = s_{min(t+K, t_g)}$; (4) train a conditional diffusion denoiser to model the dataset distribution $p(\tilde{g}|s, g)$. This process can be thought of as training a goal-conditioned behavioral cloning subgoal policy, which we refer to as the *Subgoal Diffuser*.

We do not assume the dataset contains goal-directed behavior, which has several implications: (1) $p(\tilde{g}|s, g)$ can be highly multi-modal, motivating our diffusion modeling choice. (2) $p(\tilde{g}|s, g)$ may contain diverse states in terms of value-distance from $s$ and $g$, i.e., $V(s, \tilde{g})$ and $V(\tilde{g}, g)$ respectively, making $K$ hard to set such that the subgoals are guaranteed to be *reachable* (even if we had access to $R$ a priori). (3) The subgoal Diffuser fits the behavior data and thus does not capture any notion of subgoal *optimality*. We therefore introduce a simple and effective test-time subgoal generation procedure, which we describe in the following and is summarized in Alg. 1.

We sample $N$ subgoal candidates from the subgoal diffuser and filter them for reachability based on a value threshold $\hat{R}$, i.e., keep subgoals $\tilde{g}$ that satisfy $V(s, \tilde{g}) > \hat{R}$. We then select the subgoal that is

---

**Algorithm 1** Subgoal Generation Procedure

---

**Input:** current timestep $t$, current state $s$, current subgoal $g'$, goal $g$, subgoal Diffuser $\mathcal{D}$, value function $V$, policy competence radius $\hat{R}$, subgoal samples $N$, subgoal timesteps $T_{sg}$.
**Output:** subgoal $\tilde{g}$
1: **if** $t \% T_{sg} == 0$ **then**     # Sample new subgoal
2:     Sample subgoal candidates $\{\tilde{g}_i\}_{i=1}^N \sim \mathcal{D}(s, g)$
3:     Filter reachable subgoals $\{\tilde{g}_j\}_{j=1}^M \leftarrow \{\tilde{g}_i \mid V(s, \tilde{g}_i) > \hat{R}\}$     # $M \leq N$
4:     Select subgoal closest to goal $\tilde{g} \leftarrow \arg\max_{\tilde{g} \in \{\tilde{g}_j\}} V(\tilde{g}, g)$
5:     **if** $V(\tilde{g}, g) \leq V(s, g)$ **then**
6:         $\tilde{g} \leftarrow g$     # If $s$ is closer to the goal $g$ than the generated subgoal $\tilde{g}$, go directly to $g$
7:     **end if**
8: **else**     # Keep current subgoal
9:     $\tilde{g} \leftarrow g'$
10: **end if**
11: **return** $\tilde{g}$

---

closest to the goal, i.e., with the highest value $V(\tilde{g}, g)$ (see Fig. 2, left). We rollout the low-level RL policy $\pi$ with this subgoal for a fixed $T_{sg}$ timesteps and then repeat the subgoal generation procedure. This can be viewed as a form of constrained sample-based planning with receding horizon control or Model Predictive Control (MPC), in which case our model consists of $\{\mathcal{D}, V\}$ and the optimization is performed on the state-space $\mathcal{S}$ rather than the action-space $\mathcal{A}$ (see App. D.1 for discussion). To aid convergence, if the goal is closer to the current state than the chosen subgoal it is replaced with the goal (lines 5–7 in Alg. 1). The constraint we impose on the subgoals is appropriate in our setting since $-V(s, g)$ can be viewed as a discounted (asymmetric) distance metric between states (Wang et al., 2023), and our approach is compatible with other test-time constraints on states.

### 3.3 GENERATING ENTITY-FACTORED SUBGOALS

The previous section describes a subgoal generation and selection mechanism which is agnostic to the structure of the state space. When the state of the environment can be described as a collection of entities, prior work has demonstrated that factored state representations and set-based policy architectures (e.g., Transformers (Vaswani et al., 2017)) better handle the combinatorial complexity of the state-space, yielding improved performance, greater sample efficiency, and facilitates compositional generalization (Zhou et al., 2022; Haramati et al., 2024). We further exploit this structure to generate entity-factored subgoals. In addition to the above benefits, entity-level factorization can simplify the subtask for the low-level policy by facilitating modifications to a subset of the entities. While there may be explicit ways to constrain the subgoal generation to encourage sparse modifications from the current state, we observe that this emerges naturally from the data with the appropriate inductive bias. In our case, it is the bias induced by entity-centric diffusion (Qi et al., 2025): given sets of state and goal entities $s = \{s_m\}_{m=1}^M$ and $g = \{g_m\}_{m=1}^M$, our diffusion model gradually denoises a set of noisy subgoal entities $\tilde{g}^\tau = \{\tilde{g}_m^\tau\}_{m=1}^M$, where $M$ denotes the set size and $\tau$ the diffusion timestep. Importantly, each entity in $s$, $g$ and $\tilde{g}^\tau$ is a *separate input* to the Transformer denoiser (or other set-based architecture) encoded with its affiliation to either of the sets.

As discussed in the previous section, the distribution $p(\tilde{g}|s, g)$ can encompass a number of ways to reach a given goal, which grows combinatorially with the number of entities. This leads to a wide and multi-modal distribution over the relevant subgoals. Our choice of diffusion enables capturing the multiple subgoal modes present in the data, potentially corresponding to modifying states of different subsets of entities. This in turn allows sampling from distinct modes. By contrast, producing a weighted average of those modes is more likely to result in modifications to large portions of the state (see Sec. 4.5). As our experiments show, coupling entity-centric representations with Transformer-based diffusion encourages entity-factored subgoals. We attribute this partly to the Transformer's ability to selectively copy its input tokens ($\{s_m\}_{m=1}^M$ and $\{g_m\}_{m=1}^M$) to the output ($\{\tilde{g}_m^\tau\}_{m=1}^M$) via the attention mechanism (Jelassi et al., 2024). The entity-centric structure additionally allows us to train relatively small diffusion models with as few as 10 denoising steps, which reduces computational burden for real-time control.

# 4 EXPERIMENTS

Experiments are designed to study: (1) our method's efficacy in handling long-horizon goal-conditioned tasks involving multiple entities (Sec. 4.3); (2) the impact of our design choices on performance (Sec. 4.4); (3) the quality of subgoals generated by our method (Sec. 4.5); (4) our method's compositional generalization capabilities (Sec. 4.6). We present our evaluation suite in Sec. 4.1 and describe implementation and baselines in Sec. 4.2. We provide datasets, checkpoints and code to reproduce the results in this paper at: `https://github.com/DanHrmti/HECRL`.

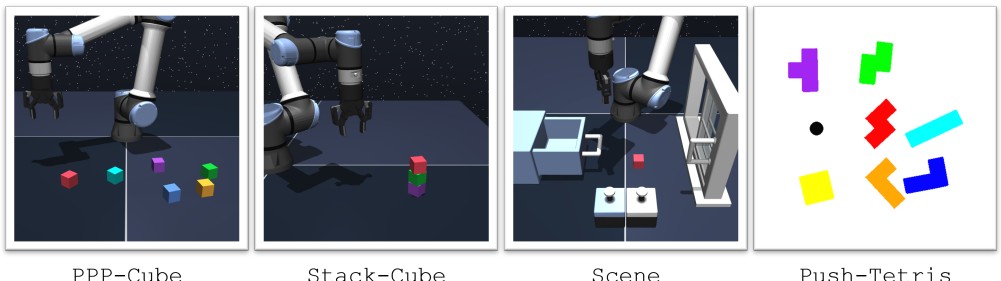

PPP-Cube      Stack-Cube      Scene      Push-Tetris

Figure 3: **Environment suite**. `PPP-Cube`: **P**ick, **P**lace and **P**ush cubes to goal positions. `Stack-Cube`: Pick-and-place cube stacking. `Scene`: Manipulate objects (drawer, window, button and cube) to goal configurations. `Push-Tetris`: Push blocks to goal positions and orientations.

## 4.1 ENVIRONMENTS, TASKS AND DATASETS

Our experimental results focus on multi-object manipulation using a suite of environments, tasks and datasets which we adapt from previous benchmarks to highlight challenges in perception and handling combinatorial state-spaces. The environments are visualized in Fig. 3. The majority of our suite builds on the OGBench (Park et al., 2025a) manipulation environments, specifically *Cube* and *Scene* environments. These contain a UR5e robot arm and involve table-top object manipulation. We increase image resolution, add multi-view perception and remove simplifying visual components such as arm transparency and end-effector coloring.

**Tasks**: (1) `PPP-Cube`: based on the OGBench *Cube* environment, the agent is required to manipulate cubes between randomly initialized state and goal positions. Distinct colors are assigned to each cube at random out of a fixed set at the beginning of each episode. The accompanying dataset contains diverse agent-object interactions including **P**ick-**P**lace and **P**ush operations. Image observations include two views, front and side. To isolate algorithmic aspects from any particular object-centric image representation, we support state-based entity-centric observations that emulate a "perfect" factored representation. (2) `Stack-Cube`: identical to `PPP-Cube` except the dataset contains only Pick-and-Place operations with a high probability of stacking and the agent is only evaluated on stacking. (3) `Scene`: We adopt the OGBench *Visual Scene* environment as is with modifications to some test tasks to avoid ambiguity in goal specification that arises due to our more realistic visual setup (see App. B). The environment is observed from a single view. (4) `Push-Tetris`: we adapt the *Push-T* environment introduced in Chi et al. (2023) to multi-object manipulation of Tetris-like blocks. Distinct block types are sampled at random at the beginning of each episode. Each block type has a distinct color which is fixed across episodes. The agent is required to push the blocks to goal configurations including position and orientation. A dataset is collected using a random policy restricted to a fixed radius around an object, sampled at fixed time intervals, resulting in highly suboptimal behavior. See App. B for extended environment details.

**Environment Characterization**: We consider `PPP-Cube` and `Stack-Cube` the most challenging in our suite[1], requiring learning long-horizon 3D manipulation in a combinatorial state-space from realistic image observations. `Scene` presents similar challenges in terms of perception and includes more object types (and corresponding manipulation capabilities) but has much fewer effective state configurations, making it significantly less combinatorially complex. `Push-Tetris` is

---

[1]see App. B.1 for a literature review highlighting that non-trivial performance on these environments with more than 2 objects has not been attained prior to this work within the data regime we consider.

more visually simple and limited to a 2D plane but requires fine-grained control of variable objects in a combinatorially complex state-space.

**Evaluation**: Training datasets contain 3 objects (excluding `Scene`) and up to $3M$ state transitions. We train agents with a fixed amount of gradient updates and report mean and standard deviation of the final checkpoint across 4 seeds for all of our experiments and metrics. Best results up to a standard deviation are highlighted in bold. In contrast to OGBench, we do not terminate upon reaching the goal at test-time such that the agent cannot just stumble upon the goal but must reach and stably maintain it. This is critical in tasks such as `Stack-Cube` where the cubes should remain stacked and `Scene` where the effective number of goal configurations allows non-trivial performance by a policy that simply explores those configurations without relying on the goal specification.

## 4.2 IMPLEMENTATION AND BASELINES

Baselines were carefully chosen to contrast the different aspects and design choices of our method (see App. Table 9) while ensuring fair comparison. The RL agents we consider are based on goal-conditioned variants of IQL (Kostrikov et al., 2022).

**EC-SGIQL (Ours)**: we implement our method on top of an ECRL (Haramati et al., 2024) agent trained with IQL. For the Subgoal Diffuser we adapt EC-Diffuser (Qi et al., 2025) to generate subgoals conditioned on entity-centric states and goals. We refer to this instantiation of our method as Entity-Centric SubGoal IQL (EC-SGIQL).

**EC-IQL**: corresponds to an ECRL (Haramati et al., 2024) agent adapted to the offline setting by integrating the Entity Interaction Transformer (EIT) architecture with IQL. This baseline represents an entity-centric non-hierarchical method, which is equivalent to ours without subgoal generation.

**EC-Diffuser** (Qi et al., 2025): an entity-centric diffusion-based behavioral cloning method.

**HIQL**: an agent based on HIQL (Park et al., 2023).

**IQL**: an agent based on IQL (Kostrikov et al., 2022).

All entity-centric methods (EC- prefix) are trained from entity-centric state observations (see App. B for details) or latent image-based representations extracted using a pretrained DLPv2 (Daniel & Tamar, 2024) (see App. A.3 for an overview of DLP). Standard agents are trained from a single-vector state observation or latent image-based representations extracted using a pretrained VQ-VAE (Van Den Oord et al., 2017). Image representations were pretrained on the offline RL datasets, one for each task (see App. D.7 for reconstruction visualizations). We provide extended implementation details in App. C.

## 4.3 LONG-HORIZON MANIPULATION

Table 1 summarizes performance of all methods on our environment suite. We report the state-goal pixel coverage (i.e., overlap) averaged over objects for `Push-Tetris` and success rates otherwise. Our method significantly outperforms all baselines with two exceptions in which it performs on par. Notably, EC-SGIQL uses the same low-level policy and value function as EC-IQL yet consistently improves its performance. On `PPP-Cube` from images, the most challenging task in our suite, it achieves more than a $150\%$ increase in success rate. EC-IQL is the second most performant method, highlighting the significance of structure in these domains. Figures 1 and 4 visualize rollouts of our method following the subgoals generated by the Diffuser in the image-based environments. See App. D.3 for further discussion and performance results and our website for rollout videos.

Table 1: **Long-horizon manipulation performance**. All values are success rates except for `Push-Tetris` for which we report state-goal pixel coverage. See Sec. 4.2 for baseline details. Best results up to a standard deviation are highlighted in **bold**.

| Environments | EC-SGIQL | EC-IQL | EC-Diffuser | HIQL | IQL |
|---|---|---|---|---|---|
| `PPP-Cube` (*State*) | **82.5** $\pm$ **3.1** | 51.5 $\pm$ 4.4 | 44.8 $\pm$ 6.7 | 48.3 $\pm$ 7.3 | 34.3 $\pm$ 4.9 |
| `Stack-Cube` (*State*) | **43.5** $\pm$ **1.9** | 29.0 $\pm$ 2.9 | **43.8** $\pm$ **9.2** | 0.0 $\pm$ 0.0 | 19.3 $\pm$ 3.0 |
| `PPP-Cube` (*Image*) | **64.3** $\pm$ **4.9** | 25.0 $\pm$ 5.7 | 0.3 $\pm$ 0.5 | 0.0 $\pm$ 0.0 | 0.0 $\pm$ 0.0 |
| `Scene` (*Image*) | **61.5** $\pm$ **5.9** | 53.0 $\pm$ 5.5 | 3.3 $\pm$ 2.5 | 8.3 $\pm$ 1.3 | 17.5 $\pm$ 2.7 |
| `Push-Tetris` (*Image*) | **61.4** $\pm$ **3.3** | 31.6 $\pm$ 1.3 | 7.9 $\pm$ 0.5 | 5.2 $\pm$ 0.8 | 3.4 $\pm$ 0.8 |

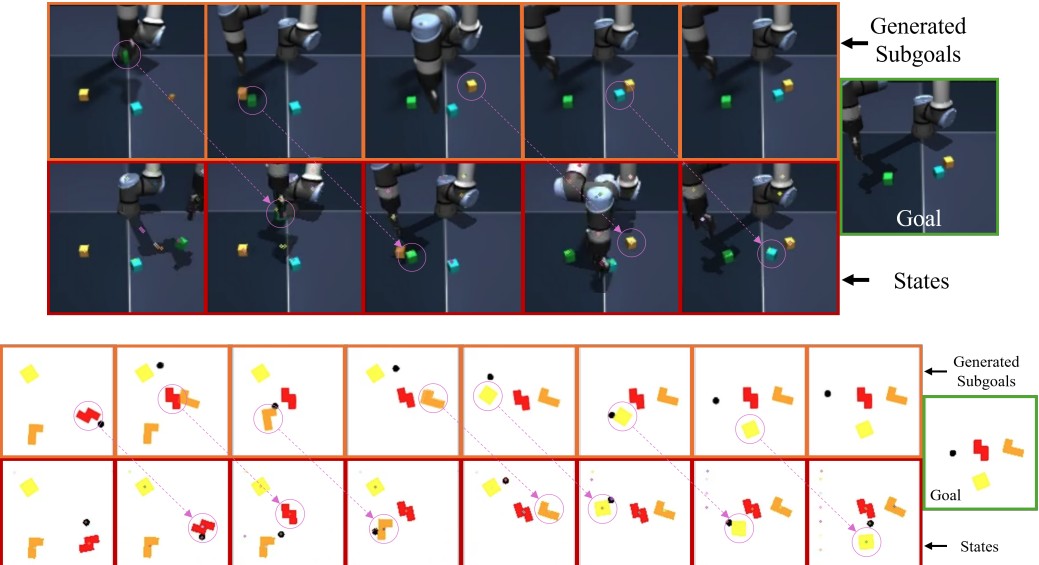

Figure 4: `PPP-Cube` and `Push-Tetris` **image-based factored subgoals**. *Bottom row*: environment image observations at the time of subgoal generation. Positions of latent particles are plotted on the image. *Top row*: DLP reconstructions of the generated latent subgoals. *Rightmost image*: goal image observation. *Circles and arrows*: circles on the top row highlight factors (excluding the agent) that are modified by the subgoals compared to the current observation in the image below them. Arrows leading to circles on the bottom row highlight the factors that are manipulated by the low-level RL policy, showing that the subgoals were achieved.

## 4.4 ABLATION STUDY

Table 2 presents results ablating the following aspects of our subgoal generation procedure: (1) use of the value function for subgoal selection (*Random Sample*), (2) use of the value function for subgoal filtering (line 3 in Alg.1) (*Max Value*) and (3) the use of diffusion by training a deterministic Transformer-based subgoal generator with an Advantage-Weighted Regression (*AWR*, (Peng et al., 2019)) objective as in HIQL. AWR with entity-centric latent image representations requires special care as the latent entities lack ordering. We adjust the standard regression loss to a loss based on the Chamfer distance (see App. C for details).

Our method consistently achieves the best results, both in terms of success rate and timestep efficiency. The *Max Value* variant performs on par on some of the tasks, indicating that the policy competence radius may be larger than the radius of the generated subgoals in these cases. Another trend we observe is that the *Random Sample* and *AWR* variants require more timesteps to reach goals when the success rates are comparable. This is expected when selecting random samples from the Diffuser which do not always make direct progress to the goal. The *AWR* variant achieves the lowest performance overall which we attribute to the quality of the subgoals. Fitting a (latent) observation-generating model with weighted regression results in weighted averages of observations, which do not generally correspond to a valid state. We study this further in the following section.

## 4.5 MEASURING SUBGOAL QUALITY

We hypothesize that our approach performs well compared to baselines and ablations in part because it encourages generating simpler subgoals with sparse changes to factors compared to the current state when the data and domain support it. We measure this sparsity in the state-based *Cube* environments and report the average number of modified entities in Table 3. Training the subgoal policy with AWR results in changes to all 3 cubes most of the time, while our Subgoal Diffuser modifies close to 1 cube on average. We attribute this to the diffusion model's ability to produce samples from distinct subgoal modes compared to the deterministic policy which produces a weighted average of those modes, as well as to the factored structure of the Subgoal Diffuser.

Table 2: **Subgoal generation ablation performance**. *Ours*: EC-SGIQL. *Max Value*: Ours w.o. filtering subgoal candidates. *Random Sample*: randomly samples a subgoal from the Diffuser. *AWR*: replaces the Diffuser with a deterministic Transformer trained with AWR. For `Push-Tetris` we report pixel coverage (top) and the sum of pixel Chamfer distances across the episode (bottom). For the rest we report success rate (top) and negative returns (bottom) which correspond to the average number of timesteps taken to either complete the task or terminate due to evaluation timestep limit. Best results up to a standard deviation are highlighted in **bold**.

| Environments | Ours | Max Value | Random Sample | AWR |
|---|---|---|---|---|
| `PPP-Cube` (*State*) | **82.5** $\pm$ **3.1** | 76.3 $\pm$ 1.0 | **73.0** $\pm$ **7.4** | 67.8 $\pm$ 5.7 |
| | **422** $\pm$ **60** | **464** $\pm$ **35** | 759 $\pm$ 23 | 773 $\pm$ 46 |
| `Stack-Cube` (*State*) | **43.5** $\pm$ **1.9** | **35.5** $\pm$ **7.5** | 28.8 $\pm$ 5.5 | 9.0 $\pm$ 4.2 |
| | **706** $\pm$ **3** | 771 $\pm$ 54 | 904 $\pm$ 21 | 985 $\pm$ 6 |
| `PPP-Cube` (*Image*) | **64.3** $\pm$ **4.9** | 46.3 $\pm$ 2.1 | **57.3** $\pm$ **8.7** | 55.5 $\pm$ 3.4 |
| | **587** $\pm$ **18** | 709 $\pm$ 7 | 750 $\pm$ 42 | 735 $\pm$ 37 |
| `Scene` (*Image*) | **61.5** $\pm$ **5.9** | **61.0** $\pm$ **6.7** | 41.0 $\pm$ 6.2 | 44.5 $\pm$ 6.0 |
| | **597** $\pm$ **26** | **589** $\pm$ **43** | 843 $\pm$ 5 | 719 $\pm$ 45 |
| `Push-Tetris` (*Image*) | **61.4** $\pm$ **3.3** | **59.5** $\pm$ **3.4** | **58.6** $\pm$ **4.5** | **56.1** $\pm$ **5.6** |
| | **510** $\pm$ **12** | **505** $\pm$ **32** | 622 $\pm$ 30 | 606 $\pm$ 40 |

Table 3: **Number of modified entities excluding the agent in the generated subgoal compared to the input state**. *EC-Diffusion*: Our subgoal Diffuser, *EC-AWR*: a Transformer trained with AWR on entity-centric states, *AWR*: an MLP trained with AWR on single-vector states. An entity is considered modified if its position changed more than a threshold distance. The environments contain 3 cubes and the robotic arm. Values are averaged over 400 randomly sampled initial states and goals. Best results up to a standard deviation are highlighted in **bold**.

| Environments | EC-Diffusion (Ours) | EC-AWR | AWR |
|---|---|---|---|
| `PPP-Cube` (*State*) | **1.36** $\pm$ **0.01** | 2.96 $\pm$ 0.01 | 2.98 $\pm$ 0.00 |
| `Stack-Cube` (*State*) | **1.04** $\pm$ **0.01** | 2.82 $\pm$ 0.02 | 2.98 $\pm$ 0.01 |

Since measuring the subgoal sparsity in the image-based environments is not straightforward, we provide qualitative observations obtained by reconstructing the latent subgoal particles with the DLP decoder for the Subgoal Diffuser in Fig. 1 and 4, and for the AWR variant in Fig. 6, 7 and 8 (App.). The Diffuser subgoals are often a composition of the input state and goal images and involve sparse changes from the state. The generated subgoals are not perfect and occasionally include the same entity twice. When this occurs, the entities are more often than not an exact copy from each of the inputs, i.e., one from the state and one from the goal. This supports our hypothesis that the Transformer selectively copies its input entities to the output. We attribute these type of errors to the nature of the DLP representation and the diffusion objective, which might be remedied with larger diffusion Transformers. The AWR subgoals often contain multiple duplicates of entities representing different futures in a *single subgoal*, which provides the low-level RL policy with ambiguous goals. See App. D.6 for further discussion.

## 4.6 GENERALIZATION

Incorporating entity-factored structure relaxes the combinatorial complexity of the state space by facilitating compositional generalization (Lin et al., 2023). Learning requires less coverage of the state space as long as there is sufficient coverage of the individual factors. Entity-level state generalization (e.g., to novel compositions of object position and color) is measured by sample efficiency. Indeed, we observe that entity-centric methods reach peak performance with much fewer gradient updates compared to the unstructured ones (when the latter achieve non-trivial performance). Compositionality with respect to the global state of the system can be tested by varying the number of state and/or goal entities. We test our method's generalization capabilities in these cases and report the performance in Table 4 (see results including `Push-Tetris` in App. Table 14). Our method

showcases non-trivial generalization which degrades with increasing number of objects. We surpass the performance of the flat entity-centric agent, showing that the subgoals maintain a level of quality sufficient to guide the RL policy. These results hint to our method's scaling potential to increasing numbers of entities via curriculum or offline-to-online finetuning, which is left for future work.

Table 4: **Zero-shot compositional generalization**. All values are success rates. * signifies the training variant in each segment, which was used to evaluate zero-shot capabilities on the other variants. Best results up to a standard deviation are highlighted in **bold**.

| Variants | EC-SGIQL | EC-IQL | EC-Diffuser |
|---|---|---|---|
| `2-Cubes-Stack` (*State*) | $69.8 \pm 7.0$ | $45.8 \pm 12.3$ | $\mathbf{87.8} \pm \mathbf{5.4}$ |
| `3-Cubes-Stack*` (*State*) | $\mathbf{43.5} \pm \mathbf{1.9}$ | $29.0 \pm 2.9$ | $\mathbf{43.8} \pm \mathbf{9.2}$ |
| `4-Cubes-Stack-2` (*State*) | $\mathbf{28.8} \pm \mathbf{3.9}$ | $17.3 \pm 2.5$ | $1.5 \pm 1.3$ |
| `PPP-1-Cube` (*State*) | $\mathbf{97.0} \pm \mathbf{1.8}$ | $92.3 \pm 8.9$ | $85.0 \pm 9.2$ |
| `PPP-2-Cubes` (*State*) | $\mathbf{94.0} \pm \mathbf{3.4}$ | $75.0 \pm 3.8$ | $76.5 \pm 7.6$ |
| `PPP-3-Cubes*` (*State*) | $\mathbf{82.5} \pm \mathbf{3.1}$ | $51.5 \pm 4.4$ | $44.8 \pm 6.7$ |
| `PPP-4-Cubes` (*State*) | $\mathbf{65.3} \pm \mathbf{3.5}$ | $31.8 \pm 4.0$ | $42.8 \pm 11.4$ |
| `PPP-5-Cubes` (*State*) | $\mathbf{49.0} \pm \mathbf{8.5}$ | $19.3 \pm 5.9$ | $23.8 \pm 6.7$ |
| `PPP-6-Cubes` (*State*) | $\mathbf{25.7} \pm \mathbf{1.5}$ | $10.5 \pm 6.1$ | $8.8 \pm 3.5$ |
| `PPP-1-Cube` (*Image*) | $\mathbf{94.5} \pm \mathbf{2.9}$ | $\mathbf{91.5} \pm \mathbf{1.3}$ | $3.8 \pm 1.3$ |
| `PPP-2-Cubes` (*Image*) | $\mathbf{77.0} \pm \mathbf{3.6}$ | $47.8 \pm 6.2$ | $0.0 \pm 0.0$ |
| `PPP-3-Cubes*` (*Image*) | $\mathbf{64.3} \pm \mathbf{6.0}$ | $25.0 \pm 5.7$ | $0.3 \pm 0.5$ |
| `PPP-4-Cubes` (*Image*) | $\mathbf{38.3} \pm \mathbf{5.7}$ | $11.5 \pm 4.7$ | $0.0 \pm 0.0$ |
| `PPP-5-Cubes` (*Image*) | $\mathbf{19.3} \pm \mathbf{6.2}$ | $4.0 \pm 2.0$ | $0.0 \pm 0.0$ |
| `PPP-6-Cubes` (*Image*) | $\mathbf{9.5} \pm \mathbf{1.3}$ | $1.5 \pm 1.3$ | $0.0 \pm 0.0$ |

## 5 CONCLUSION

We present Hierarchical Entity-Centric Reinforcement Learning (HECRL), an offline GCRL framework that integrates subgoal hierarchy with factored structure to solve long-horizon tasks in multi-entity domains and scales to image observations. We design our method to be simple, flexible and modular, making it compatible with various value-based GCRL algorithms and test-time subgoal constraints. We empirically demonstrate that our factored Subgoal Diffuser coupled with our simple value-based selection procedure produces high-quality subgoals to guide a low-level entity-centric RL agent, consistently boosting its performance. We study what aspects contribute to the performance of our method and find that the bias induced by entity-centric diffusion encourages subgoals with sparse modifications from the current state compared to commonly used weighted regression objectives. Finally, we display non-trivial zero-shot generalization performance with increasing number of entities (and consequently longer horizons), hinting to potential in scaling to environments with more entities.

**Limitations and Future Work**: our method assumes that the value function provides a sufficient signal for the low-level policy to have a non-negligible competence radius as well as for guiding the subgoal generator, which held in our challenging domains but may limit its applicability in others. That said, the modularity of our framework facilitates improvements on top of advances in flat value-based GCRL algorithms. The Subgoal Diffuser is trained with subgoals up to a fixed $K$ steps away from the current state. While our method is more robust to this hyperparameter due to our test-time subgoal filtering mechanism (see App. D.5), future work can explore ways to automatically infer $K$ from data. Finally, applying our method to domains with image observations relies on a good factored state estimator. We show that this is possible in our simulated domains without requiring supervision. The potential of scaling our approach to real-world in-the-wild scenarios thus goes hand in hand with advancements in unsupervised object-centric representation learning (Seitzer et al., 2023; Zadaianchuk et al., 2023; Daniel & Tamar, 2024).

## ETHICS STATEMENT

This work adheres to the ICLR Code of Ethics. Our work advances methods for offline Goal-Conditioned Reinforcement Learning (GCRL). The experiments in this paper were conducted in simulated environments, with no involvement of human subjects or privacy concerns, and we do not foresee any ethical or societal risks from this work.

## REPRODUCIBILITY STATEMENT

We are committed to enabling full reproducibility of our work. We have included extensive implementation and evaluation details throughout the paper and in the Appendix. We release code, checkpoints and datasets to reproduce all of the results in this paper and facilitate future research at: https://github.com/DanHrmti/HECRL.

## ACKNOWLEDGMENTS

This research was supported by the ONR under grant N00014-22-1-2592.
AZ and CQ were funded by NSF 2340651, NSF 2402650, DARPA HR00112490431, and ARO W911NF-24-1-0193. AT received funding from the European Union (ERC, Bayes-RL, Project Number 101041250). Views and opinions expressed are however those of the authors only and do not necessarily reflect those of the European Union or the European Research Council Executive Agency. Neither the European Union nor the granting authority can be held responsible for them.

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

APPENDIX

THE USE OF LARGE LANGUAGE MODELS (LLMS)

LLMs were used solely to polish writing in some parts of the paper.

## A EXTENDED BACKGROUND & RELATED WORK

### A.1 MULTI-ENTITY ENVIRONMENTS

Slightly more formally, we define a *multi-entity environment* as one that can be characterized by a factored state-space: $\mathcal{S} = \mathcal{S}_1 \otimes \mathcal{S}_2 \otimes ... \otimes \mathcal{S}_N$, where some attributes are shared across the different factors $\{S_i\}_{i=1}^N$ which constitute the entities. One example is robotic multi-object manipulation where the robot and objects are the entities, each described by shared attributes such as position and visual appearance (and potentially others which are not shared). Another example is robotic locomotion, where each robot joint can be viewed as a separate entity described by shared physical properties. A comprehensive review of structure in (deep) reinforcement learning that defines and discusses different forms of state factorization can be found in Mohan et al. (2024).

### A.2 OBJECT-CENTRIC REPRESENTATIONS FOR SEQUENTIAL DECISION-MAKING

Many previous studies have dealt with the question of how to leverage the factored structure in environments with multiple entities for sequential decision-making. Some assume access to the underlying factored state of the system and develop set-based architectures to process them in a sample-efficient and generalizing manner both in model-free (Li et al., 2020; Mambelli et al., 2022; Zadaianchuk et al., 2022; Zhou et al., 2022) and model-based (Sanchez-Gonzalez et al., 2018; Sancaktar et al., 2022) settings. In order to scale these methods to image observations, some methods take an end-to-end learning approach by learning an object-centric representation concurrently with the decision-making modules (Zambaldi et al., 2019; Watters et al., 2019; Veerapaneni et al., 2020; Ferraro et al., 2025). Others leverage unsupervised Object-Centric Representations (OCRs) including patch-based (Jiang* et al., 2020), slot-based (Locatello et al., 2020) and particle-based (Daniel & Tamar, 2022) models. These are commonly pretrained and kept frozen for downstream training of the decision-making components. These have been applied in both model-free (Zadaianchuk et al., 2021; Yoon et al., 2023; Haramati et al., 2024), model-based (Zhao et al., 2022; Chang et al., 2023; Mosbach et al., 2024; Zhang et al., 2025) and imitation learning (Qi et al., 2025) approaches.

### A.3 DEEP LATENT PARTICLES

This section provides an overview of the Deep Latent Particles (DLP, Daniel & Tamar (2022; 2024)) model which we employ as the unsupervised object-centric image representation in our experiments. DLP is a Variational Auto-Encoder (VAE, Kingma & Welling (2013)) with a structured latent space consisting of a set of latent vectors referred to as *particles*. Particles encode local regions in the image that ideally correspond to salient factors such as objects or parts of objects. Each particle representation is comprised of the following attributes: pixel-space 2D position $z_p \in \mathbb{R}^2$, scale of the box bounding the region it represents $z_s \in \mathbb{R}^2$, "depth" attribute used to model occlusion between particles $z_d \in \mathbb{R}$, transparency $z_t \in \mathbb{R}$ and visual latent features that encode the appearance of the particle region $z_v \in \mathbb{R}^n$, $n$ denoting the per-particle visual latent dimension hyperparameter. A separate particle is allocated to encode the background of the image. The number of particles $M$ is a hyperparameter—which has been chosen in previous work as well as this one—to upper-bound the number of entities of interest in the image. It is worth noting that DLP is entirely unsupervised and it is not guaranteed that each particle represent an object in the image nor that an object will be represented by a single particle. Allocating a large number of particles increases the likelihood that all entities of interest are captured by the representation with the price of increased dimensionality and computational complexity both for DLP training and downstream decision-making. See Figures 9, 10 and 11 for DLP decompositions of images from the environments we use in our experiments.

## B  ENVIRONMENTS, TASKS AND DATASETS

We provide detailed descriptions of the environments, tasks and datasets we use in this work for transparency and reproducibility purposes. We highlight the differences from the environments they are based on for the reader's convenience. We introduce novel variations of tasks from the OGBench benchmark (Park et al., 2025a) implemented with the MuJoco (Todorov et al., 2012) simulator and a novel variant of the *Push-T* environment introduced in Chi et al. (2023). OGBench implementations are based on the official code found in: `https://github.com/seohongpark/ogbench`). We base our implementation of *PushT* on a variant introduced in Zhou et al. (2025) using the paper's official code found in: `https://github.com/gaoyuezhou/dino_wm`. We hope that these new variants, which highlight challenges less studied in popular benchmarks such as combinatorial state-spaces and compositional generalization, will facilitate further research in the field of entity-centric decision-making and beyond. Datasets and code including environment implementations and data collection scripts can be found at `https://sites.google.com/view/hecrl`. See our website for video demonstrations of the data collection policies.

`PPP-Cube`
***Environment***: based on the OGBench *Cube* environment, the agent is required to manipulate cubes between randomly initialized state and goal positions in a 3D space. Distinct colors are assigned to each cube at random out of a fixed set of 6 colors at the beginning of each episode. The modified multi-color setup effectively increases the size of the state-space and allows testing generalization to increasing number of objects, and consequently longer horizons, without requiring generalization to new colors. Image observations include two views, front and side. To isolate algorithmic aspects from any particular object-centric image representation, we support state-based entity-centric observations that emulate a "perfect" factored representation. Specifically, each entity's (i.e., robot gripper and cubes) state is represented by its 3D position and yaw angle. The robot arm's state includes gripper opening and the cube state is padded with 0 to match the robot's state dimension. Each entity's state is concatenated to a 1-hot identifier $e_i \in \{0, 1\}^7$. $e_0$ represents the agent and the rest represent the different colored cubes. Pixel observations are multi-view RGB images of dimension $128 \times 128$. Action space dimensionality is 5 and includes deltas in gripper 3D position, yaw and opening. A goal is considered reached when all cubes are within a threshold distance from the specified position.
***Dataset***: the dataset contains diverse agent-object interactions including **P**ick, **P**lace and **P**ush operations collected with a noisy scripted policy. The policy randomly selects a cube to either: (1) pick and place in a random location or (2) push from a random face. The dataset contains 3 cubes and a total of 3M transitions, 7500 episodes each with 400 transitions.
***Evaluation***: Maximum evaluation episode length for 3 cubes is 1000 timesteps. We evaluate each agent with a checkpoint trained for 2.5M gradient steps on 100 randomly sampled initial state and goal configurations.
***Differences from Original***: (1) Pixel observations are multi-view RGB images of dimension $128 \times 128$ compared to OGBench which are single-view RGB images of dimension $64 \times 64$. (2) OGBench modifies the visual properties of the robot arm to be semi-transparent highlight the end-effector with purple color, which we remove in our version. (3) State observations are minimal compared to OGbench and do not contain robot joint positions and velocities, gripper contact and object quaternions. (4) Each cube in our environments can be in 1 of 6 colors (without repetition per instantiation) regardless of the number of cubes where in OGBench these are fixed and depend on the number of cubes in the environment. (5) The dataset collection policy in our setting is different, the major difference being the push operation. (6) Episode length in our dataset is 400 compared to 1000 in OGBench.

`Stack-Cube`
***Environment***: see `PPP-Cube`.
***Dataset***: the dataset contains Pick-and-Place operations with a high probability of stacking collected by a noisy scripted policy. The scripted policy includes a recovery mechanism in cases it is in the processes of stacking a block and moves any of the lower blocks in the stack, which involves placing those blocks back in their position before attempting to stack the original one on top of them. This makes the dataset much more goal-directed. We found this to be crucial for learning robust RL policies that can achieve non-trivial performance on 3-cube stacking (especially in our evaluation setup that does not terminate upon reaching the goal). To the best of our knowledge, since the release of the OGBench benchmark, no method has achieved non-trivial performance on 3-cube stacking

with the OGBench dataset. Our dataset contains 3 cubes and a total of 3M transitions, 3000 episodes each with 1000 transitions.

***Evaluation***: Maximum evaluation episode length for 3 cubes is 1000 timesteps. We evaluate each agent with a checkpoint trained for 3M gradient steps on 200 randomly sampled initial state and goal configurations.

***Differences from Original***: see (1)–(4) in `PPP-Cube`. (5) The dataset collection policy in our setting is different as described above.

`Scene`

***Environment***: We adopt the OGBench *Visual Scene* environment that involves manipulation of various object types (button, cube, drawer, window) in a 3D space. We make modifications to some test tasks to avoid ambiguity in goal specification that arises due to our more realistic visual setup where the robot arm is not transparent. Specifically, we modify tasks 4 and 5 in which goals involve images with the cube inside a closed drawer. In these cases, the agent cannot infer the desired location of the cube since it may be occluded. The goal is instead to place the cube in the drawer when it is opened and locked. Pixel observations are single-view RGB images of dimension $128 \times 128$. Action space dimensionality is 5 and includes deltas in gripper 3D position, yaw and opening. A goal is considered reached when all objects are within a threshold distance from the specified position. In Table 5 we detail what each task involves, denoting subtask sequential dependencies with $\rightarrow$ and non-sequential subtasks with $|$.

***Dataset***: Data collection policy is identical to that of OGBench. A noisy scripted policy performs subtasks in random order based on their applicability at the given state (e.g., putting the cube in the drawer will only be selected if the drawer is open). The dataset contains a total of 1M transitions, 1000 episodes each with 1000 transitions.

***Evaluation***: Maximum evaluation episode length is 1000 timesteps. We evaluate each agent with a checkpoint trained for 1.5M gradient steps on 50 randomly perturbed initial state and goal configurations per task.

***Differences from Original***: see (1)–(2) in `PPP-Cube`. (3) We slightly modify task 4 and 5 for reasons described above.

`Push-Tetris`

***Environment***: we adapt the *Push-T* environment introduced in Chi et al. (2023) to multi-object manipulation of Tetris-like blocks. Distinct block types are sampled at random at the beginning of each episode. Each block type has a distinct color which is fixed across episodes. The agent is required to push the blocks to goal configurations including position and orientation in a 2D space. Pixel observations are single-view RGB images of dimension $128 \times 128$. Action space dimensionality is 2 and includes deltas in agent 2D position. A goal is considered reached when all objects reach over $85\%$ state-goal pixel coverage. Obtaining this coverage requires very fine manipulation capabilities. We see that in practice the agent is able to learn from the sparse reward but achieves low success rates at test-time although end states are visually similar to the goal. We therefore report coverage as the main metric rather than success rate since it is more informative. Maximum evaluation episode length for 3 objects is 1000 timesteps.

***Dataset***: A dataset is collected using a random policy restricted to a fixed radius around an object, sampled at fixed time intervals, resulting in highly suboptimal behavior.

***Evaluation***: Maximum evaluation episode length for 3 objects is 1000 timesteps. We evaluate each agent with a checkpoint trained for 1M gradient steps on 100 randomly sampled initial state and goal configurations.

***Differences from Original***: (1) The original T-block is replaced with 7 different tetris-like blocks. (2) Our dataset contains semi-random interaction with objects in contrast to the original dataset that contains expert demonstrations. (3) The goal is specified by a separate image in our setting compared to a shaded region in the same image as the state observation in the original task.

Table 5: **Detailed** `Scene` **task descriptions**. We denote subtask sequential dependencies with $\rightarrow$ and non-sequential subtasks with $|$.

| Task | Subtasks | Total Subtasks | Longest Dependency |
|---|---|---|---|
| 1 | open-drawer $\|$ open-window | 2 | 1 |
| 2 | unlock-drawer $\rightarrow$ close-drawer $\rightarrow$ lock-drawer $\|$ unlock-window $\rightarrow$ close-window $\rightarrow$ lock-window | 6 | 3 |
| 3 | open-drawer $\|$ unlock-window $\rightarrow$ close-window $\|$ move-cube-to-side | 4 | 2 |
| 4 | open-drawer $\rightarrow$ lock-drawer & place-cube-in-drawer | 3 | 2 |
| 5 | unlock-drawer $\rightarrow$ open-drawer $\rightarrow$ lock-drawer & place-cube-in-drawer $\|$ unlock-window $\rightarrow$ open-window $\rightarrow$ lock-window | 7 | 3 |

## B.1 ENVIRONMENT DEGREE OF DIFFICULTY

We highlight the complexity of the domains we consider in this work and the efficacy of our method in solving them by comparing to concurrent work that requires orders of magnitude more data on similar or more simple variants of the OGBench (Park et al., 2025a) *Cube* domains. Li et al. (2025) require $100M$ transitions in a $4$-cube environment to learn in a single-task RL (i.e., not goal-conditioned) setting while learning from states and semi-sparse rewards (rewarding per-object success) and allowing additional online interaction. We exhibit non-trivial zero-shot generalization performance in manipulating $4$ cubes in a fully offline goal-conditioned sparse-reward setting while learning from only $3M$ transitions containing 3 cubes, both from states and images. Park et al. (2025b) show that unstructured RL methods, including hierarchical ones, require up to $1B$ transitions in a similar setting to obtain non-trivial performance with more than 2 objects. To the best of our knowledge, based on a review of all papers that have cited the OGBench benchmark at the time of writing this paper, there are no other methods that have attained non-trivial performance on the *Cube* domain with more than 2 objects thus far.

## C   METHOD IMPLEMENTATION DETAILS

In this section we provide extensive implementation details for our method, baselines and image representations used in our experiments. Table 9 compares methods with respect to various algorithmic aspects.

We implement all RL methods in pytorch based on official implementations and LeanRL: `https://github.com/meta-pytorch/LeanRL`, a more efficient version of CleanRL (Huang et al., 2022). The RL agents (i.e., all of the methods excluding EC-Diffuser) are based on goal-conditioned variants of IQL (Kostrikov et al., 2022). Low-level policies are extracted from IQL Q-functions with DDPG+BC (Fujimoto & Gu, 2021). Value function training goals are sampled uniformly from future states in the same trajectory with $0.8$ probability and otherwise taken as the next state. Low-level policy training goals are sampled uniformly from future states in the same trajectory (except for in HIQL). We report shared hyperparameters in Table 6 and environment-method-specific DDPG+BC policy extraction coefficient $\alpha$ in Table 7.

Table 6: **Shared hyperparameters**.

| Hyperparameter | Value |
|---|---|
| Batch size | 512 |
| Learning rate | 0.0003 |
| Gradient clip norm | 20 |
| Discount factor $\gamma$ | 0.99 |
| Target smoothing coefficient $\tau$ | 0.005 |
| IQL/HIQL expectile | 0.9 |
| AWR temperature $\beta$ | 3.0 |
| Subgoal Diffuser diffusion steps | 10 |
| Subgoal $K$ | 50 |
| EIT attention dimension | 64 |
| EIT attention heads | 8 |
| EIT hidden dimension | 256 |
| MLP layers | 4 |
| MLP hidden dimension | 512 |

**EC-SGIQL (Ours)**: we implement our method on top of an ECRL (Haramati et al., 2024) agent trained with IQL (see following EC-IQL for details). For the Subgoal Diffuser we adapt EC-Diffuser (Qi et al., 2025) to generate subgoals conditioned on entity-centric states and goals. Specifically, we remove action inputs entirely and condition on initial state and goal entities as clean inputs which are not denoised. We use learned additive embeddings to encode each individual entity's affiliation to either of the 3 sets (i.e., state, goal or noisy subgoal) and one of 2 views in the multi-view setting. The Diffuser architecture is an 8-layer Transformer which is conditioned on the diffusion timestep via adaptive layer normalization (AdaLN). Environment-specific Diffuser

Table 7: **DDPG+BC policy extraction coefficient** $\alpha$.

| Environment | EC-SGIQL (Ours) | Ours w. AWR | EC-IQL | HIQL | IQL |
|---|---|---|---|---|---|
| PPP-Cube (*State*) | 0.1 | 0.1 | 0.1 | 0.1 | 0.1 |
| PPP-Cube (*Image*) | 0.2 | 0.2 | 0.2 | 0.3 | 0.2 |
| Stack-Cube (*State*) | 0.05 | 0.05 | 0.05 | 0.2 | 0.05 |
| Scene (*Image*) | 0.3 | 0.3 | 0.3 | 0.4 | 0.2 |
| Push-Tetris (*Image*) | 0.1 | 0.1 | 0.1 | 0.1 | 0.1 |

test-time hyperparameters are detailed in Table 8. For our implementation we adapt code from the official EC-Diffuser repository: `https://github.com/carl-qi/EC-Diffuser`.

Table 8: **Subgoal Diffuser test-time hyperparameters**.

| Environment | Diffusion Samples $N$ | Subgoal Steps $T_{sg}$ | Competence Radius $\hat{R}$ |
|---|---|---|---|
| PPP-Cube (*State*) | 256 | 25 | $-30$ |
| PPP-Cube (*Image*) | 256 | 25 | $-30$ |
| Stack-Cube (*State*) | 256 | 25 | $-30$ |
| Scene (*Image*) | 64 | 50 | $-25$ |
| Push-Tetris (*Image*) | 64 | 25 | $-20$ |

**EC-SGIQL with AWR**: This method is an ablation of our method and is thus identical to it except for the subgoal policy which is a Transformer based on the Entity Interaction Transformer (EIT) (Haramati et al., 2024) which replaces the final aggregation attention with another self attention block. It is trained with an AWR objective using the IQL value function. Since the output and target are unordered entities when learning from DLP particles, we use the Chamfer distance as the loss instead of the standard MSE.

**EC-IQL**: corresponds to an ECRL (Haramati et al., 2024) agent adapted to the offline setting by integrating the Entity Interaction Transformer (EIT) architecture with IQL. We slightly modify the EIT architecture for the Q-function by replacing the action entity conditioning with AdaLN conditioning. We base our implementation on the official ECRL repository: `https://github.com/DanHrmti/ECRL`.

**HIQL**: this baseline corresponds to an agent based on HIQL (Park et al., 2023). We adapt the official implementation to make it similar to the other methods for a fair comparison while keeping the core attributes unchanged. Specifically, we train it with an underlying IQL agent and extract the low-level policy with DDPG+BC. Contrary to the other IQL-based methods in this work, as in the original HIQL, we train the low-level policy on the subgoal distribution rather than the goal distribution. The high-level policy is trained with AWR using the IQL value function. At test-time, the subgoal is kept for the same duration as the Diffuser subgoals (see Table 8). All agent components are 4-layer MLPs with a hidden dimension of 512. We base our implementation on the official OGBench implementation at: `https://github.com/seohongpark/ogbench`.

**IQL**: this baseline corresponds to a goal-conditioned agent based on IQL (Kostrikov et al., 2022). All agent components are 4-layer MLPs with a hidden dimension of 512. We base our implementation on the official OGBench implementation at: `https://github.com/seohongpark/ogbench`.

**EC-Diffuser** (Qi et al., 2025): is an entity-centric diffusion-based behavioral cloning method. It employs a Transformer-based architecture that takes as input the current state and a goal state and predicts future states and actions via diffusion. For control, it predicts future states and actions jointly, and executes the first denoised action in an MPC fashion. For all of our tasks, we use a prediction horizon of 5 and 10 diffusion steps. We use the reported value for all other hyperparameters. We use the official implementation from: `https://github.com/carl-qi/EC-Diffuser`.

Table 9: **Comparison of algorithmic aspects across different methods**.

| Attribute | EC-SGIQL (Ours) | Ours w. AWR | EC-IQL | EC-Diffuser | HIQL | IQL |
|---|---|---|---|---|---|---|
| *RL* | ✓ | ✓ | ✓ | ✗ | ✓ | ✓ |
| *Entity-centric* | ✓ | ✓ | ✓ | ✓ | ✗ | ✗ |
| *Hierarchical* | ✓ | ✓ | ✗ | ✗ | ✓ | ✗ |
| *Obs. Diffusion* | ✓ | ✗ | ✗ | ✓ | ✗ | ✗ |

**Image Representations**: We provide hyperparameters for DLPv2 (Daniel & Tamar, 2024) and VQ-VAE (Van Den Oord et al., 2017) models in Tables 10 and 11 respectively. We use the implementations provided in: `https://github.com/DanHrmti/ECRL`.

Table 10: **DLP hyperparameters**.

| Environment | Prior Particles | Posterior Particles | Visual Feature Dim | BG Particle Dim |
|---|---|---|---|---|
| `PPP-Cube` (*Image*) | 32 (per view) | 20 (per view) | 8 | 1 |
| `Scene` (*Image*) | 32 | 24 | 8 | 1 |
| `Push-Tetris` (*Image*) | 32 | 20 | 8 | 1 |

Table 11: **VQ-VAE hyperparameters**.

| Hyperparameter | Value |
|---|---|
| Embedding dimension | 16 |
| Dictionary size | 2048 |
| Flattened latent dimension | 1024 |

# D  ADDITIONAL RESULTS AND DISCUSSION

## D.1  WORLD MODEL PERSPECTIVE

Our approach has strong connections to model-based RL (Sutton, 1991) and world modeling (Ha & Schmidhuber, 2018). Whereas standard world models model the next-state distribution given the current state and action $p\left(s_{t+1}|s_t, a_t\right)$, our approach models the multi-step next-state distribution given the current state and desired goal $p\left(s_{t+K}|s_t, s_g\right)$. This modeling choice facilitates long-horizon goal-reaching by planning in the temporally-extended state space rather than the low-level action space. This in turn allows longer look-ahead planning horizons ($K = 50$ in our experiments) compared to autoregressive world models (typically around $K = 5$) which suffer from compounding model errors. Compared to diffusion world models (i.e., diffusers), we focus the modeling capacity on accurately capturing single distant states rather than multiple immediate state-action pairs. This enables generation of high quality subgoals which can be fed to the low-level RL policy although it was never trained on the diffusion-generated subgoal states, only on dataset states.

## D.2  THE BENEFITS OF DECOUPLING TRAINING GOAL DISTRIBUTIONS ACROSS THE HIERARCHY

In our goal-conditioned setting, the value function can be viewed as a discounted (asymmetric) distance metric between states (Wang et al., 2023). The compounding error and discount factor define an "effective radius" of values that provide a clear learning signal for policy extraction, which we refer to as the *policy competence radius*, $R$ (see Sec. 3.1). This quantity is not known apriori because we do not have access to the true value approximation error, and it is not clear if it can be recovered or effectively approximated in practice without evaluating an extracted policy. Previous hierarchical methods spanning offline RL (Park et al., 2023) and diffusion planning (Li et al., 2023; Chen et al., 2024) fix a single hyperparameter $K$ defining the training distributions for both levels of the hierarchy: the low-level policy is trained on state-goal pairs that are at most $K$ timesteps apart while

the high-level policy is trained to produce these $K$-step states as subgoals. Selecting $K$ represents a tradeoff between complexity and sample efficiency: small $K$ simplifies the policy goal distribution but makes less use of the offline data by limiting the state-goal pairs the policies are trained on, potentially hindering generalization. Selecting $K$ thus requires simultaneously balancing this tradeoff across all levels of the hierarchy.

We opt for an alternative approach: train the low-level policy on the full goal distribution, i.e., the same distribution the value function is trained on. This choice decouples the training goal dependency between the hierarchies and allows us to implicitly select $K$ for the low-level policy by tuning a hyperparameter $\hat{R}$ as a test-time constraint (details in Sec. 3.2), which is preferable to tuning $K$ since it does not require re-training. $K$ can then be chosen more flexibly for the subgoal generator to upper-bound the low-level policy's competence $R$ while balancing the complexity tradeoff.

### D.3 Long-horizon Object Manipulation

Intermediate success (e.g., the number of subtasks achieved) is reported in Table 12 and detailed results on `Scene` tasks in Table 13. EC-IQL performs on par with our method on most tasks in `Scene` and is the second leading method in terms of performance overall. We believe this is due to the entity-centric structure which helps learn accurate, internally factored value functions. `Scene` is the least combinatorially complex which may explain why the hierarchy provides less benefit: the factored value function provides a good learning signal in this domain. EC-Diffuser performs on par with our method on `Stack-Cube`. We believe this is mostly due to the nature of the dataset which is more goal-directed compared to the others, and the fact that it does not rely on a value function and thus does not suffer from the same long-horizon challenges as the RL methods. HIQL and IQL do not scale well to image-based tasks with multiple objects when learning from single-vector representations, which is consistent with previous findings in the object-centric decision-making literature as well as the results on the OGBench benchmark (Park et al., 2025a). HIQL even underperforms IQL in some cases, which we believe is due to the fact that training the subgoal policy with AWR often results in low quality subgoals (see Sec. 4.4 and D.6).

Table 12: **Long-horizon manipulation performance**. All values are success fractions except for `Push-Tetris` for which we report state-goal pixel Chamfer distance. Best results up to a standard deviation are highlighted in **bold**.

| Environments | EC-SGIQL | EC-IQL | EC-Diffuser | HIQL | IQL |
|---|---|---|---|---|---|
| PPP-Cube (*State*) | **88.0** $\pm$ **1.4** | 60.8 $\pm$ 4.5 | 66.5 $\pm$ 7.0 | 62.8 $\pm$ 6.5 | 46.5 $\pm$ 7.0 |
| PPP-Cube (*Image*) | **82.3** $\pm$ **2.2** | 60.8 $\pm$ 1.5 | 9.5 $\pm$ 1.3 | 0.0 $\pm$ 0.0 | 0.0 $\pm$ 0.0 |
| Stack-Cube (*State*) | 58.8 $\pm$ 1.7 | 43.0 $\pm$ 4.6 | **55.3** $\pm$ **6.9** | 6.0 $\pm$ 0.8 | 38.5 $\pm$ 1.0 |
| Scene (*Image*) | **90.0** $\pm$ **2.2** | 84.0 $\pm$ 2.7 | 51.5 $\pm$ 0.6 | 65.3 $\pm$ 1.7 | 77.3 $\pm$ 0.5 |
| Push-Tetris (*Image*) | 19.0 $\pm$ 3.5 | 54.7 $\pm$ 4.9 | 84.2 $\pm$ 3.6 | 94.1 $\pm$ 0.9 | **98.8** $\pm$ **1.8** |

Table 13: **Detailed success rates for `Scene` tasks**. Best results up to a standard deviation are highlighted in **bold**.

| Task | EC-SGIQL | EC-IQL | EC-Diffuser | HIQL | IQL |
|---|---|---|---|---|---|
| 1 | **85.0** $\pm$ **6.0** | **83.5** $\pm$ **4.4** | 14.0 $\pm$ 14.3 | 20.0 $\pm$ 5.2 | 27.0 $\pm$ 4.8 |
| 2 | 83.5 $\pm$ 1.9 | **89.0** $\pm$ **2.0** | 6.0 $\pm$ 3.3 | 15.0 $\pm$ 4.8 | 30.5 $\pm$ 16.7 |
| 3 | **52.0** $\pm$ **7.1** | 16.5 $\pm$ 13.4 | 0.0 $\pm$ 0.0 | 7.0 $\pm$ 2.6 | 15.0 $\pm$ 4.8 |
| 4 | **43.0** $\pm$ **10.5** | **39.0** $\pm$ **15.9** | 0.5 $\pm$ 1.0 | 0.5 $\pm$ 1.0 | 6.0 $\pm$ 1.6 |
| 5 | **44.0** $\pm$ **17.2** | 37.0 $\pm$ 7.8 | 0.0 $\pm$ 0.0 | 0.0 $\pm$ 0.0 | 10.0 $\pm$ 7.1 |

### D.4 Generalization

Compositional generalization results of the entity-centric methods we compare in this work are presented in Table 14. Maximum evaluation episode length in the generalization experiments with 4, 5, 6 and 7 objects is 1200, 1500, 2000 and 2000 respectively. We additionally increase the number of inference particles—a feature supported by DLP—to 24, 28, 30 and 30 respectively.

Table 14: **Zero-shot compositional generalization**. All values are success rates except for `Push-Tetris` variants for which we report state-goal pixel coverage. * signifies the training variant in each segment, which was used to evaluate zero-shot capabilities on the other variants. Best results up to a standard deviation are highlighted in **bold**.

| Variants | EC-SGIQL | EC-IQL | EC-Diffuser |
|---|---|---|---|
| `2-Cubes-Stack` (*State*) | $69.8 \pm 7.0$ | $45.8 \pm 12.3$ | $\mathbf{87.8} \pm \mathbf{5.4}$ |
| `3-Cubes-Stack*` (*State*) | $\mathbf{43.5} \pm \mathbf{1.9}$ | $29.0 \pm 2.9$ | $\mathbf{43.8} \pm \mathbf{9.2}$ |
| `4-Cubes-Stack-2` (*State*) | $\mathbf{28.8} \pm \mathbf{3.9}$ | $17.3 \pm 2.5$ | $1.5 \pm 1.3$ |
| `PPP-1-Cube` (*State*) | $\mathbf{97.0} \pm \mathbf{1.8}$ | $\mathbf{92.3} \pm \mathbf{8.9}$ | $85.0 \pm 9.2$ |
| `PPP-2-Cubes` (*State*) | $\mathbf{94.0} \pm \mathbf{3.4}$ | $75.0 \pm 3.8$ | $76.5 \pm 7.6$ |
| `PPP-3-Cubes*` (*State*) | $\mathbf{82.5} \pm \mathbf{3.1}$ | $51.5 \pm 4.4$ | $44.8 \pm 6.7$ |
| `PPP-4-Cubes` (*State*) | $\mathbf{65.3} \pm \mathbf{3.5}$ | $31.8 \pm 4.0$ | $42.8 \pm 11.4$ |
| `PPP-5-Cubes` (*State*) | $\mathbf{49.0} \pm \mathbf{8.5}$ | $19.3 \pm 5.9$ | $23.8 \pm 6.7$ |
| `PPP-6-Cubes` (*State*) | $\mathbf{25.7} \pm \mathbf{1.5}$ | $10.5 \pm 6.1$ | $8.8 \pm 3.5$ |
| `PPP-1-Cube` (*Image*) | $\mathbf{94.5} \pm \mathbf{2.9}$ | $\mathbf{91.5} \pm \mathbf{1.3}$ | $3.8 \pm 1.3$ |
| `PPP-2-Cubes` (*Image*) | $\mathbf{77.0} \pm \mathbf{3.6}$ | $47.8 \pm 6.2$ | $0.0 \pm 0.0$ |
| `PPP-3-Cubes*` (*Image*) | $\mathbf{64.3} \pm \mathbf{6.0}$ | $25.0 \pm 5.7$ | $0.3 \pm 0.5$ |
| `PPP-4-Cubes` (*Image*) | $\mathbf{38.3} \pm \mathbf{5.7}$ | $11.5 \pm 4.7$ | $0.0 \pm 0.0$ |
| `PPP-5-Cubes` (*Image*) | $\mathbf{19.3} \pm \mathbf{6.2}$ | $4.0 \pm 2.0$ | $0.0 \pm 0.0$ |
| `PPP-6-Cubes` (*Image*) | $\mathbf{9.5} \pm \mathbf{1.3}$ | $1.5 \pm 1.3$ | $0.0 \pm 0.0$ |
| `Push-Tetris-1-Object` (*Image*) | $\mathbf{8.0} \pm \mathbf{1.5}$ | $\mathbf{6.6} \pm \mathbf{2.6}$ | $5.0 \pm 1.8$ |
| `Push-Tetris-2-Objects` (*Image*) | $\mathbf{61.4} \pm \mathbf{5.7}$ | $40.8 \pm 4.2$ | $6.3 \pm 0.4$ |
| `Push-Tetris-3-Objects*` (*Image*) | $\mathbf{61.4} \pm \mathbf{3.3}$ | $31.6 \pm 1.3$ | $7.9 \pm 0.5$ |
| `Push-Tetris-4-Objects` (*Image*) | $\mathbf{52.4} \pm \mathbf{3.5}$ | $20.7 \pm 1.7$ | $6.4 \pm 1.1$ |
| `Push-Tetris-5-Objects` (*Image*) | $\mathbf{41.8} \pm \mathbf{6.0}$ | $14.4 \pm 2.3$ | $4.6 \pm 0.6$ |
| `Push-Tetris-6-Objects` (*Image*) | $\mathbf{32.0} \pm \mathbf{1.8}$ | $11.5 \pm 2.2$ | $4.6 \pm 1.2$ |
| `Push-Tetris-7-Objects` (*Image*) | $\mathbf{21.1} \pm \mathbf{2.2}$ | $10.0 \pm 1.5$ | $4.7 \pm 0.8$ |

### D.5 SUBGOAL DIFFUSER HYPERPARAMETER SENSITIVITY STUDY

We test the sensitivity of our method to the subgoal-related hyperparameters $K$, $T_{sg}$ and $N$ on `PPP-Cube` (*Image*), the most challenging task in our suite. Results are presented in Fig. 5. We find that the method is largely robust to these hyperparameters, significantly outperforming the underlying RL agent with all of the tested configurations.

$K$ (Fig. 5, left): results align with our intuition that filtering subgoals for reachability makes our method robust to $K$ as long as it upper-bounds the policy competence radius $R$ (see discussion in App. D.2). Performance degrades when $K = 10$, hinting that the competence radius is larger than this value.

$T_{sg}$ (Fig. 5, center): Larger values incur faster average action execution time since they require less diffusion generations but may result in decreased decision-timestep efficiency if the value is much larger than what the RL policy requires to reach the subgoal. In addition, given that the sampled subgoals are not perfect, regenerating subgoals at a reasonable rate can benefit the agent. This is analogous to the re-planning performed in MPC. We see that for larger values, performance begins to degrade.

$N$ (Fig. 5, right): small sample sizes decrease the probability of sampling "optimal" subgoals which results in high variance in performance. Large sample sizes increase the probability of generating erroneously high-valued subgoals which can degrade performance. Moderate sample sizes between $64 - 256$ result in equivalently high success rates.

### D.6 AWR SUBGOAL VISUALIZATION

We provide visualizations of subgoals generated by a deterministic entity-centric Transformer trained with AWR on DLP representations in Figures 6, 7 and 8. We observe that across all environments, subgoals often contain duplications of entities, providing ambiguous subgoals for the low-level GCRL policy. This duplication can be explained by the redundancy in the DLP represen-

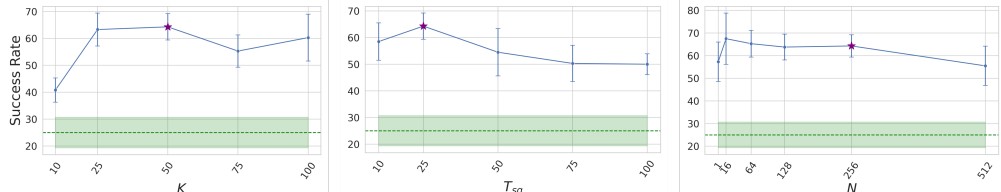

Figure 5: **Hyperparameter sensitivity study**. Success rates of our method on `PPP-Cube` (*Image*). A single hyperparameter is varied in each plot while the others are unchanged compared to the ones used for the main results, which we refer to as *default*. Purple stars - *default values*, error bars - *standard deviations*, dashed lines and shaded regions in green - *mean and standard deviation of the EC-IQL baseline without subgoals*. **Left**: varying $K$, the Subgoal Diffuser training distribution timestep difference between state and subgoal. **Center**: varying $T_{sg}$, the test-time subgoal rollout timesteps. **Right**: varying $N$, the number of test-time diffusion subgoals sampled before filtering.

tation and the AWR objective. As discussed in Sec. 4.5, weighted regression fits a weighted average of future observations, which in our "play" datasets, contain many possible futures given an initial state and a goal. DLP represents a scene with many particles, which due to its unsupervised nature, may represent the same object with multiple particles (see App. A.3). This allows each particle to capture a different future subgoal for the *same object* in a *single subgoal*, which is what we observe in these figures. The subgoal averaging phenomenon is not purely an artifact of the redundancy in the DLP representation given that it occurs in the state-based environments as well (see Sec. 4.5, Table 3). These results highlight the multi-modality in the data and help explain the lower performance of baselines and ablations using AWR to train the subgoal generator. This phenomenon rarely happens with our Subgoal Diffuser (and is much less severe when it does), which captures the multiple modes in the data and enables separately sampling from them at test-time. We believe that larger diffusion Transformers may help mitigate this phenomenon entirely.

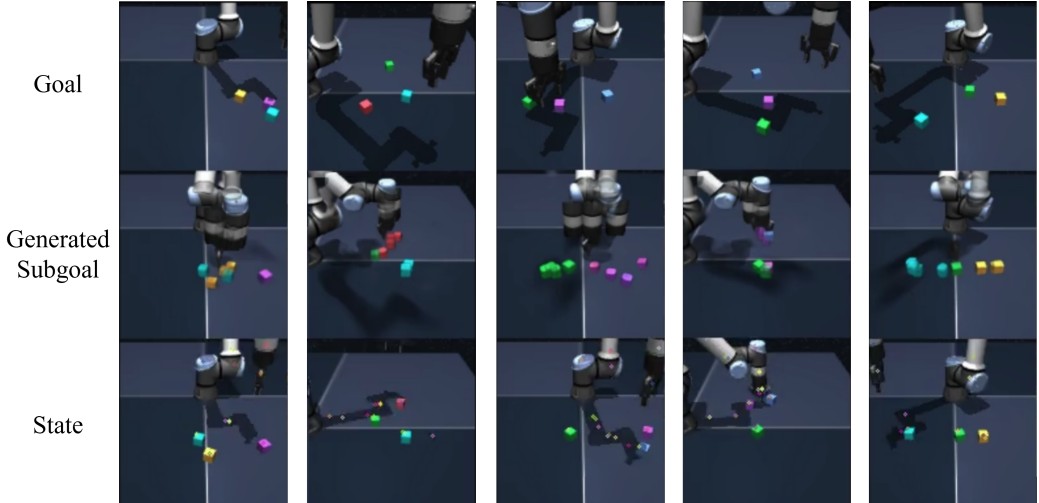

Figure 6: `PPP-Cube` **AWR subgoals**. Subgoals (middle) are reconstructed with the DLP decoder and were generated conditioned on DLP representations of the state (bottom) and goal (top). Columns are not sequential, i.e., each column represents unrelated subgoals.

## D.7 PRETRAINED IMAGE REPRESENTATION RECONSTRUCTION

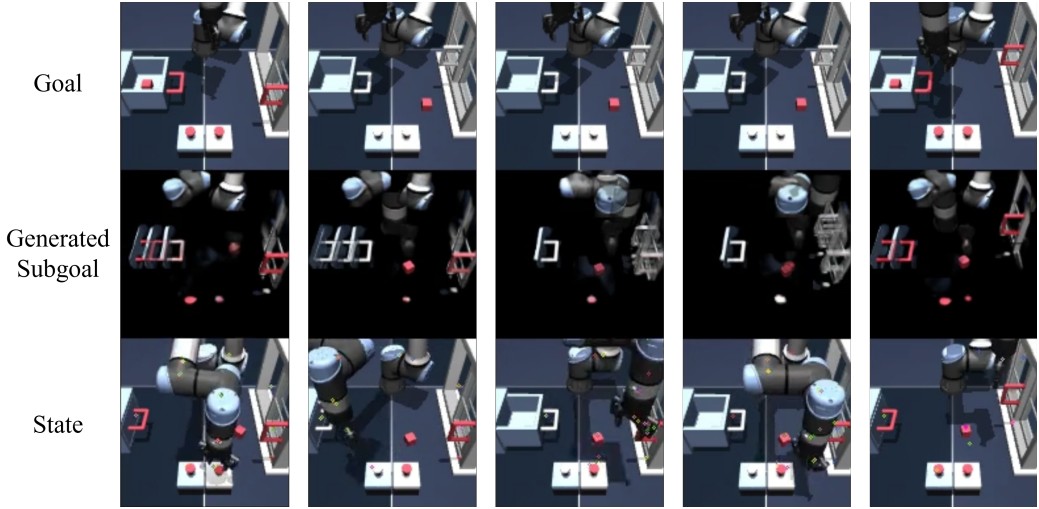

Figure 7: `Scene` **AWR subgoals**. Subgoals (middle) are reconstructed with the DLP decoder and were generated conditioned on DLP representations of the state (bottom) and goal (top). Columns are not sequential, i.e., each column represents unrelated subgoals.

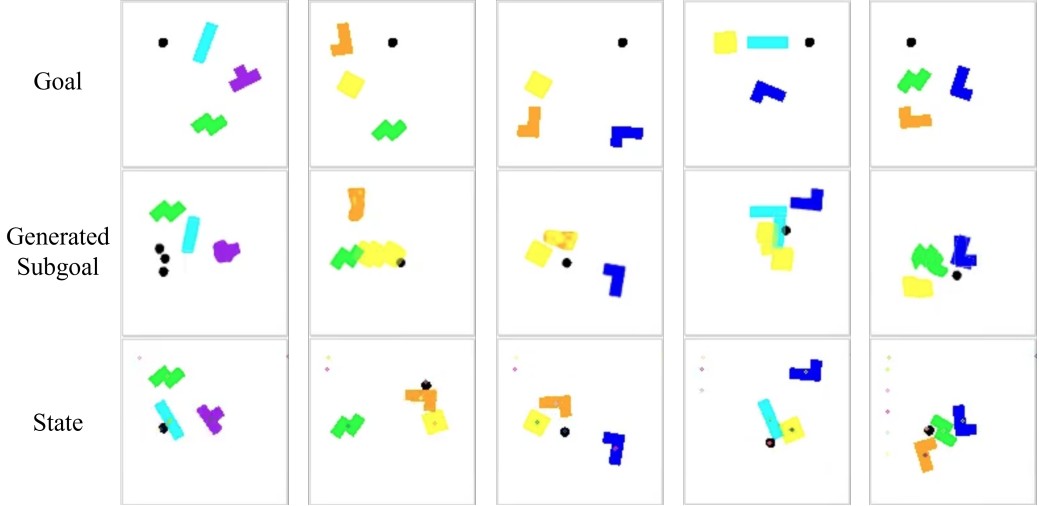

Figure 8: `Push-Tetris` **AWR subgoals**. Subgoals (middle) are reconstructed with the DLP decoder and were generated conditioned on DLP representations of the state (bottom) and goal (top). Columns are not sequential, i.e., each column represents unrelated subgoals.

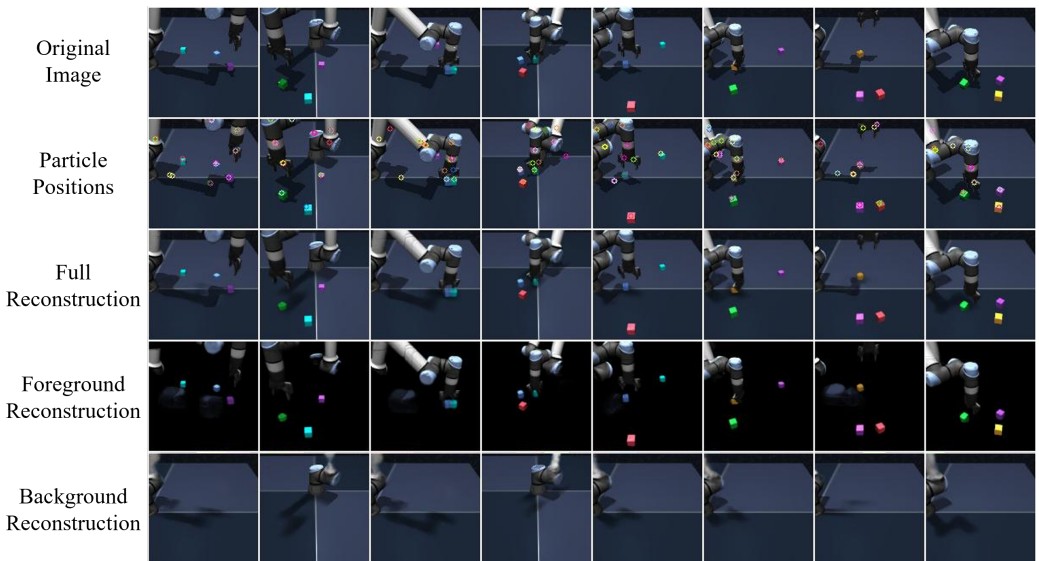

Figure 9: `PPP-Cube` **DLP decomposition**.

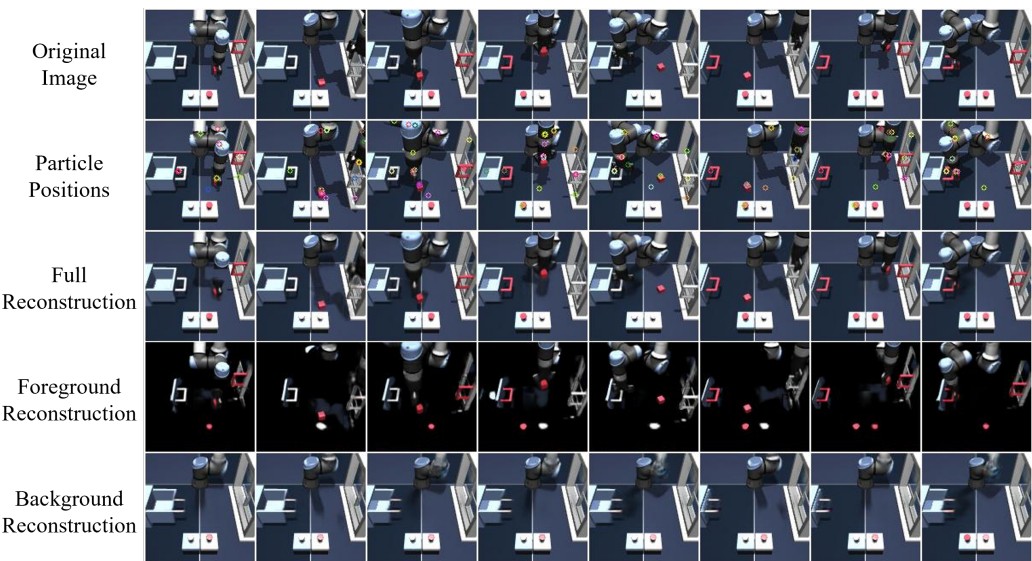

Figure 10: `Scene` **DLP decomposition**.

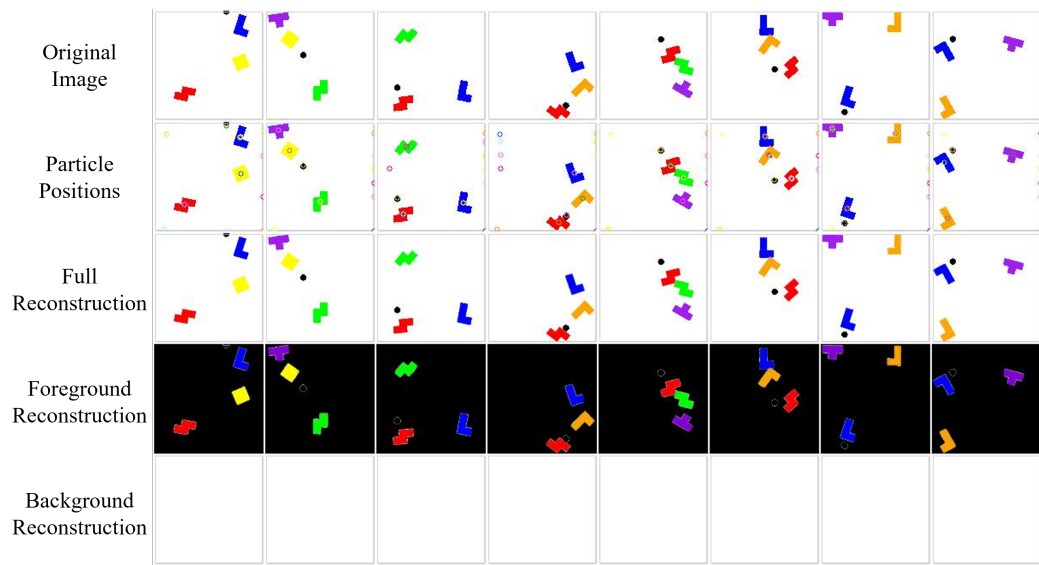

Figure 11: `Push-Tetris` **DLP decomposition**.

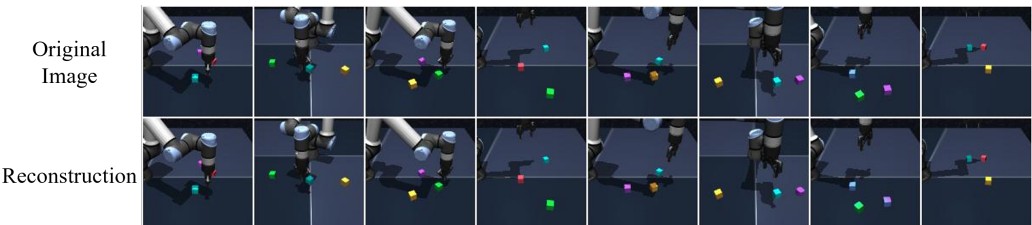

Figure 12: `PPP-Cube` **VQ-VAE reconstruction**.

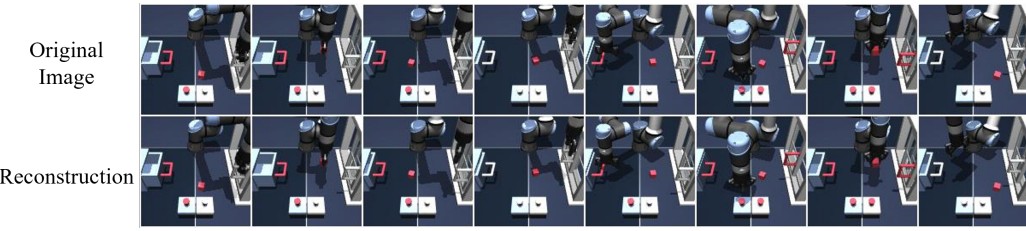

Figure 13: `Scene` **VQ-VAE reconstruction**.

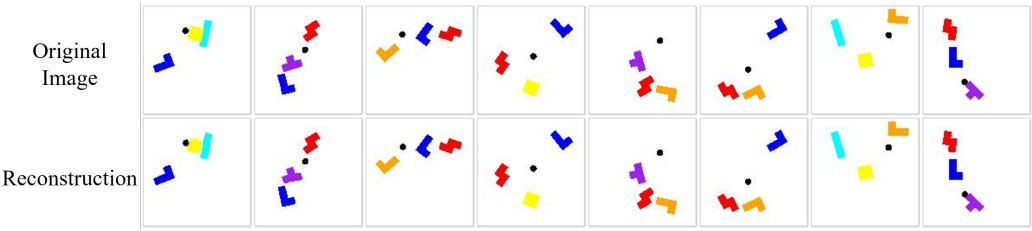

Figure 14: `Push-Tetris` **VQ-VAE reconstruction**.

