# OpenReview forum: "Hierarchical Entity-centric Reinforcement Learning with Factored Subgoal Diffusion"
_ICLR.cc/2026/Conference — ICLR 2026 Poster_

### Official Review · Reviewer_pQXN · 2025-10-27

**Soundness:** 2
**Presentation:** 3
**Contribution:** 3
**Rating:** 6
**Confidence:** 2

**Summary:**

This paper has proposed a novel hierarchical reinforcement learning approach, where the subgoals are entity-centric. In the proposed approach, the policies are implemented with the diffusion networks. The experiment results demonstrate that the proposed approach significantly outperforms the state-of-the-art hierarchical baselines. Beyond that, the entity-centric subgoals improve the explainability of the learned policies.

**Strengths:**

1.	The hierarchical learning problem is of great importance in the RL domain, which is promising for solving the long-horizon problems in real-world applications.

2.	The proposed approach is extensively evaluated in the simulated robotic manipulation domain and has demonstrated impressive experimental performance.

**Weaknesses:**

1.	Figure 2 is confusing. The goals are represented with small squares. It is unclear what the difference is between the factored squares and the unfactored squares, and even how they influence the learning process.

2.	The authors are encouraged to show more about the learned subgoals, and the goal-conditioned policies. It is unclear what goal-conditioned policies are learned which leads to the superior task performance.

**Questions:**

1.	What is the difference between entities and objects?
2.	Does the paper assume that the collection of entities in Line 129 is given before learning?

---

> ### Author Response · Authors · 2025-11-17
>
> We appreciate the acknowledgment of our method’s novelty, strong performance and explainability compared to previous state-of-the-art methods.
>
> We thank you for your questions and constructive remarks, which we address in the following.
>
> *Figure 2 is confusing. The goals are represented with small squares. It is unclear what the difference is between the factored squares and the unfactored squares, and even how they influence the learning process.*
>
> Thank you for pointing this out. We have edited the caption to highlight the difference between the two types of subgoals in the new revision, please let us know if the clarity is improved.
> We would also like to clarify that the factored subgoals are *a result of our training process* and positively *influence test-time performance*. As we mention throughout the paper, (entity-)factored subgoals in the context of our experiments refers to “subgoals with sparse changes to entities compared to the current state”. We show that by leveraging factored subgoals, our method improves over prior subgoal generation methods since they are more easily reachable by the low-level policy, requiring manipulation of a single object (on average) per subgoal.
>
> *The authors are encouraged to show more about the learned subgoals, and the goal-conditioned policies. It is unclear what goal-conditioned policies are learned which leads to the superior task performance.*
>
> We summarize the analysis we have provided on the *learned subgoals* other than the improved performance compared to baselines:
> - We present quantitative results on the "factoredness" of the produced subgoals in the state-based environments in Table 3.
> - We present qualitative verbal descriptions of our observations of the generated subgoals in the second paragraph of Section 4.5, and further discuss and compare to baselines in Appendix D.5.
> - We present qualitative image visualizations of the generated subgoals in Figures 1, 5 and 6 (and for baselines in Figures 7, 8 and 9).
> - We present video visualizations of the subgoals generated sequentially (middle image of each video) as the low-level policy reaches them (bottom image of each video), on our website (link in abstract).
>
> Regarding the underlying *goal-conditioned RL policies*, our framework builds on existing low-level GCRL policies. Specifically, we implement an adaptation of [ECRL](https://arxiv.org/abs/2404.01220) to the offline RL setting in the form of EC-IQL (which did not previously exist), which also serves as a baseline. The video visualizations on our website present rollouts of EC-IQL guided by the Subgoal Diffuser’s generated subgoals (which is the instantiation of our method used in the experiments), and demonstrate the learned underlying goal-conditioned policies.
>
> We are happy to add other metrics/analysis/visualizations the reviewer finds relevant.
>
> *What is the difference between entities and objects?*
>
> Thank you for this important question. We believe that neither “object” nor “entity” is well defined without context. The term “object” is normally associated with human notions of objects. Since we are dealing with unsupervised image representations in this work, we choose the term “entity”  following previous work in the field since we find it more general.
>
> In our OGBench Cube experiments for example, the state-based entity factorization aligns well with what a human would refer to as an “object” (cubes, robot arm) but the image-based entity factorization obtained with Deep Latent Particles captures different parts of the robot arm with separate “particles", and often allocates more than a single “particle” to the same cube.
>
> *Does the paper assume that the collection of entities in Line 129 is given before learning?*
>
> We do not assume this. The entity factorization in the state-based environments is naturally given since the attributes of each entity (agent and objects) are maintained separately in simulation. Obtaining an entity-factorization from images is not trivial, but we show that a useful factored representation can be obtained without any supervision using unsupervised object-centric representation learning. In our experiments we use the Deep Latent Particles representation, which we provide further details on in Appendix A.3.

---

### Official Review · Reviewer_K6LA · 2025-10-31

**Soundness:** 2
**Presentation:** 3
**Contribution:** 2
**Rating:** 4
**Confidence:** 4

**Summary:**

This paper proposes an offline Goal-Conditioned Reinforcement Learning (GCRL) framework to enhance learning efficiency and generalization in long-horizon goal-reaching tasks within complex environments featuring multiple entities. The authors employ a hierarchical structure comprising a diffusion model for generating entity-factored subgoals and a value-based GCRL agent.

**Strengths:**

While hierarchical structures have been explored in prior work, utilizing diffusion models to generate subgoals that incorporate information from multiple entities stands out as a novel approach. Additionally, modifying the experimental environments to highlight the advantages of the proposed method and comparing it against baselines strengthens the paper's contributions.

**Weaknesses:**

The paper would benefit from more detailed explanations of model design choices, along with ablation studies to justify them. Although the use of diffusion for subgoal generation is innovative, it remains unclear whether this component is essential for the performance gains. There is a concern that filtering subgoals—generated via the unused value function during diffusion model training—might play a more critical role than the diffusion process itself.

**Questions:**

1. As mentioned in the Weakness section, an ablation study on subgoal filtering using value functions would be helpful. It would be useful to compare scenarios where subgoals generated by diffusion are used directly, or where multiple subgoal candidates are randomly sampled and then filtered, against the current selection method.
2. I am curious about the impact of hyperparameters K and N, as described in Section 3.2, on performance.
3. The algorithm indicates that the current subgoal g' is an input, but this is not explicitly addressed in the description. What role does the current subgoal play?
4. Is it feasible to compare against a baseline that directly applies the EC-IQL method to HIQL?
5. According to Table 1, HIQL achieves a 0 success rate in PPP-Cube (image) and Stack-Cube (state). What explains this lower performance compared to the cube environment in OGBench? Is it due to the increased number of cubes that need to be manipulated?
6. Why is only the Stack-Cube (state) environment included, while the Stack-Cube (image) version is omitted?

---

> ### Author Response · Authors · 2025-11-17
>
> We appreciate the acknowledgment of the novelty of our approach and the contribution of our modified environment suite.
>
> We additionally thank you for the detailed and actionable review and questions, which we address in the following.
>
> *As mentioned in the Weakness section, an ablation study on subgoal filtering using value functions would be helpful. It would be useful to compare scenarios where subgoals generated by diffusion are used directly, or where multiple subgoal candidates are randomly sampled and then filtered, against the current selection method.*
>
> We would like to kindly point out that the requested ablation has already been conducted and presented in Section 4.4 with performance results in Table 2. We would also like to clarify that our selection method randomly samples multiple subgoal candidates (Alg. 1 line 2) and then filters them based on our criteria (Alg. 1 lines 2-3). For convenience, we summarize our findings below:
>
> - We compare our subgoal generation strategy with using the subgoals generated by diffusion directly (Random Sample column in Table 2) and find that our strategy outperforms it both in terms of success rate and timestep efficiency, highlighting the efficacy of our approach in extracting high-quality directed subgoals from the diffusion model distribution.
>
> - We additionally evaluate our subgoal generation strategy without filtering for reachability (Max Value column in Table 2) and show that the reachability constraint has significant positive effects on performance.
>
> *I am curious about the impact of hyperparameters K and N, as described in Section 3.2, on performance.*
>
> We agree that this will be a valuable addition to the paper. We have conducted this study and found that our method is largely robust to these hyperparameters, significantly outperforming the underlying RL agent with all of the tested configurations. We provide a detailed study with full results in Appendix D.4 in the new revision.
>
> *The algorithm indicates that the current subgoal g' is an input, but this is not explicitly addressed in the description. What role does the current subgoal play?*
>
> We appreciate your attention to detail. We believe the confusion is due to a typo in Alg.1 line 9, which we have corrected in the new revision. The role of the current subgoal is solely for line 9 in the algorithm, i.e., maintaining the same subgoal for an extended time horizon $T_{sg}$.
>
> *Is it feasible to compare against a baseline that directly applies the EC-IQL method to HIQL?*
>
> It is feasible and it is essentially what the ablation variant AWR (Table 2, rightmost column) represents, i.e., we have implemented the baseline you suggest and compared our method with it. We would like to highlight that this integration (applying EC-IQL to HIQL) is not trivial and required novel algorithmic components including adapting the [EIT](https://arxiv.org/abs/2404.01220) architecture to produce subgoals as well as modifying the standard regression loss to a loss based on the Chamfer distance for compatibility with the entity-factored set representation. We found that modeling the subgoals using a deterministic Transformer trained with AWR is significantly outperformed by our method, and further study and identify the cause to be low-quality subgoals (see Section 4.5 and Appendix D.5).

---

> > ### Author Response · Authors · 2025-11-17
> > **response continued**
> >
> > *According to Table 1, HIQL achieves a 0 success rate in PPP-Cube (image) and Stack-Cube (state). What explains this lower performance compared to the cube environment in OGBench? Is it due to the increased number of cubes that need to be manipulated?*
> >
> > The most direct comparison between HIQL’s performance in the setting of our paper and the OGBench setting (although there are various differences highlighted throughout the paper) is HIQL’s performance on the *cube-triple-noisy-v0* and *visual-cube-triple-noisy-v0* environments. HIQL’s results in our paper are consistent or better than the results in the [OGBench paper](https://arxiv.org/abs/2410.20092):
> >
> > - *Manipulation of 3 cubes from states without stacking*: Table 18 in the OGBench paper shows that HIQL achieves 8% success rate on task 1 which requires manipulating only a single cube, and 0% on the other tasks. In our setting on the other hand, HIQL achieves ~50% success rate.
> > - *Stacking 3 cubes from states*: Table 18 task 5 in OGBench corresponds to 3-cube stacking, on which HIQL achieves 0% success rate which is consistent with our results.
> > - *Manipulation of 3 cubes from images*: Table 21 shows that HIQL achieves 99% success rate on task 1 (which requires manipulating only a single cube), 2% success rate on task 2, 14% success rate on task 3 (with a standard deviation of 11%) and 0% otherwise, which is practically ~0% success rate on 3-cube manipulation and consistent with our results.
> >
> > We find that non-factored approaches struggle with the combinatorial complexity of multi-object manipulation environments, especially when learning from images, which is consistent with previous work in the field and highlights the efficacy of our approach in solving such problems.
> >
> > *Why is only the Stack-Cube (state) environment included, while the Stack-Cube (image) version is omitted?*
> >
> > Our method requires that the underlying RL policy achieves non-negligible performance such that it can at least reach nearby subgoals. As our results on Stack-Cube (state) suggest, this is a very difficult task (lowest success rate out of all of the tasks including image-based tasks). Since we were not yet able to obtain non-trivial success rates with EC-IQL on this task from images, we did not include it in our evaluation suite.
> >
> > We would also like to highlight the fact that we are the first to demonstrate non-trivial performance  (i.e., >> 0% success rate) on these environments since they were first introduced by the OGBench benchmark at the time of submission (see Appendix B.1 for further details.)

---

> > > ### Comment · Reviewer_K6LA · 2025-11-27
> > >
> > > Thank you for the detailed response. The modifications made during the rebuttal period, along with your responses, have resolved all the questions I had. I will therefore update my score accordingly.

---

### Official Review · Reviewer_q4dk · 2025-10-31

**Soundness:** 3
**Presentation:** 3
**Contribution:** 3
**Rating:** 6
**Confidence:** 3

**Summary:**

This paper presents Hierarchical Entity-Centric RL (HECRL), composed of a value-based GCRL and a factored subgoal-generating diffusion model to address long-horizon tasks in domains with multiple entities. In addition, the authors introduce a new version of benchmark problems that highlight the challenges of entity-centric tasks.

**Strengths:**

This paper is well written, well-constructed and easy to follow. In particular, its pictorial illustration, such as Figure 2, was very helpful in understanding the overall idea. The proposed method effectively addressed challenging long-horizon multiple-entity tasks.

**Weaknesses:**

The proposed components are based on existing methodologies, such as entity-factored subgoals and subgoal diffuser, so the novelty of the approach itself may be limited. However, the meaningful combination of these components has yielded strong performance. Please see others in questions.

**Questions:**

**Questions**

Q. What is the major difference between the proposed model and EC-diffuser?

Q. What is the key difference from the previously proposed method, which utilizes a diffusion model to generate a subgoal? Is it modularity that enables test-time subgoal selection with the value function?

Q. Sensitivity study with respect to $K$ and $T_{sg}$ is missing.

Q. What is the meaning of boldface numbers in Table 2? There are multiple numbers of boldface numbers. The essential explanation seems to be missing or at least not highlighted enough to check.

**Minor Comments**

C. An explicit definition of the entity-centric environment would be helpful.

C. In Figure 1, DLP is presented without a proper definition. Please at least use Deep Latent Particles (DLP) \cite{xx}.

---

> ### Author Response · Authors · 2025-11-17
>
> We appreciate the recognition of the quality of writing, presentation clarity and demonstration of our framework’s strong performance.
>
> We would also like to thank you for your constructive feedback and questions which are clearly aimed to improve and further understand our work. We address your questions in the following.
>
> *What is the major difference between the proposed model and EC-diffuser?*
>
> There are several main differences between our framework and EC-Diffuser. For a comprehensive response, we will begin by stating the main commonality between the two—the architecture, which we adopt from EC-Diffuser with slight modifications and repurpose for our Subgoal Diffuser.
>
> The main differences are:
> 1. *The learned distribution*: EC-Diffuser is an imitation learning method trained to model the expert dataset distribution of state-action trajectories of horizon $H$ given an initial state $s_0$ and goal $g$, $p(a_0, s_1, a_1, \dots, a_{H-1}, s_H | s_0, g)$. Our Subgoal Diffuser is trained to model a suboptimal “play” dataset distribution of single $K$-step states given an initial state $s_0$ and goal $g$, $p(s_K | s_0, g)$. $K$ is typically much larger than $H$ ($K=50$ in our experiments compared to $H=5$ in EC-Diffuser), which points to a significant algorithmic choice—in order to solve long horizon tasks we focus the modeling capacity on accurately capturing single distant states rather than multiple immediate state-action pairs. This enables generation of high quality subgoals which can be explicitly fed to the low-level RL policy although it was never trained on the diffusion-generated subgoal states, only on dataset states.
> 2. *Model usage*: EC-Diffuser generates a single output trajectory per state-goal pair and uses it at test-time for Model Predictive Control (MPC) by executing action $a_0$ and then regenerating a trajectory from the resulting state (similar to the original Diffuser work). Important to this comparison, it does not use the generated states explicitly in any way. In our method, the Subgoal Diffuser generates multiple subgoal state candidates per state-goal pair (e.g., $N=256$), out of which one is selected to be given to the low-level RL policy to reach for an extended timestep period $T_sg$. This difference is essential since we do not just use the Diffuser to capture the multi-modal distribution but explicitly leverage it by sampling from multiple modes simultaneously and selecting the best sample based on our criteria.
>
> To summarize, while we adopt the EC-Diffuser architecture almost as is (we discard the action tokens), we repurpose it in a new setting and novel manner in terms of the learned distribution and its role in sequential decision-making.
>
> *What is the key difference from the previously proposed method, which utilizes a diffusion model to generate a subgoal? Is it modularity that enables test-time subgoal selection with the value function?*
>
> Can you please clarify which “previously proposed method” you are referring to? If referring to EC-Diffuser, please see our answer to your previous question above. EC-Diffuser does not generate subgoals but a trajectory of states that are not used explicitly but serve as an auxiliary objective that improves downstream performance. As for previous work using diffusion models to generate subgoals (such as the hierarchical diffusers mentioned in the related work section), the key difference is that our Subgoal Diffuser generates a single $K$-step subgoal compared to a trajectory of $K$-step subgoals. A key insight that our paper may bring to the wider diffusion-based decision-making community is that the role of diffusion models should be considered carefully with respect to performance and complexity. Our results suggest that one may reduce the complexity of diffuser training and inference by modeling single states in the appropriate timestep difference rather than full trajectories, and rely more on the value function for long-horizon planning.
>
> The modularity and test-time flexibility is what distinguishes our approach from previous hierarchical RL methods such as HIQL which train a deterministic subgoal policy and couples the training distributions of the policies. We provide a discussion on this matter in Appendix D.1.

---

> > ### Author Response · Authors · 2025-11-17
> > **response continued**
> >
> > *Sensitivity study with respect to $K$ and $T_{sg}$*
> >
> > We agree that this study will be a valuable addition to the paper. We have conducted this study and found that our method is largely robust to these hyperparameters, significantly outperforming the underlying RL agent with all of the tested configurations. We provide a detailed study with full results in Appendix D.4 in the new revision.
> >
> > *What is the meaning of boldface numbers in Table 2? There are multiple numbers of boldface numbers. The essential explanation seems to be missing or at least not highlighted enough to check.*
> >
> > The boldface numbers denote the best results up to a standard deviation, which means there can be more than a single value highlighted in each row if the means $+-$ std overlap. This is stated in section 4.1 under “Evaluation” and is the standard practice we are familiar with. We have added a sentence in each of the relevant tables that highlights the meaning of boldface numbers. Specifically in Table 2 each row within an environment row presents a different performance metric which is described in the caption.
> >
> > *An explicit definition of the entity-centric environment would be helpful.*
> >
> > Thank you for this remark. We would like to highlight that “entity-centric” better describes the approach we take rather than the environments/domains. A more appropriate name would be “multi-entity” or “entity-factored” environments. We have replaced the term “entity-centric environment/domain” with “multi-entity environment/domain” in the new revision. We have also added a more explicit description of the characteristics of such an environment in Appendix A.1 in the new revision.
> >
> > *In Figure 1, DLP is presented without a proper definition.*
> >
> > We appreciate you pointing this out. We have modified the caption to include the full name and citation, and referenced the Appendix for a detailed description.

---

> > > ### Comment · Reviewer_q4dk · 2025-11-25
> > > **Thank you**
> > >
> > > Thank you for the effort and response, especially for the additional sensitivity analysis. The explanation helps me further understand the novelty and effectiveness of this work. I have no further inquiries and would like to maintain my current score.

---

### Official Review · Reviewer_m4Xk · 2025-11-10

**Soundness:** 3
**Presentation:** 2
**Contribution:** 3
**Rating:** 6
**Confidence:** 4

**Summary:**

This work proposes a two part hierarchical reinforcement learning framework that targets environments with multiple entities and sparse reward. Their framework (HECRL) is composed of 1. Subgoal diffuser: a high-level diffusion model that generates immediate subgoal, and 2. a value based Goal Conditioned Reinforcement Learning agent that does test-time subgoal selection.

The work tackles non-stationarity and instability in off-policy hierarchical RL (the issue of reward error propagation in long horizon tasks) by producing immediate subgoal that is reachable and provides clear learning signal to the policy learning agent. This is done by enforcing a value based reachability constraint at test time.

Experiments on several continuous-control benchmarks (Reacher, Pusher, Point Maze, Ant Maze variants, Ant Fall) show improved sample efficiency and final success rates over strong HRL baselines such as HIRO, HRAC, HIGL, SAGA, and HLPS, as well as smaller gaps between generated and actually reached subgoals, and ablations indicate that both diffusion and GP components contribute to the gains

**Strengths:**

1. Originality: The framework is modular, compatible with various value-based GCRL algorithms. Existing hierarchical diffusers diffuse over global subgoals without explicit entity factorization. EC-Diffuser uses entity-centric diffusion but for behavior cloning rather than subgoal generation. Hence, the proposed combination of conditional diffusion model over entity-factored subgoals with a value-based GCRL agent  is novel. The paper also make it clear that their framework builds directly on two mature lines of work — hierarchical diffusion for subgoals and entity-centric RL, and that their contribution is meaningful but compositional, rather than a completely new paradigm.

2. Significance: The contribution is more than a minor architectural tweak. Using GP prior over the high-level mapping as a separate surrogate distribution provides explicit predictive means and variances over subgoals: this defines a hybrid selection rule that mixes diffusion-sampled subgoals with the GP mean. The paper also shows theoretical justification beyond heuristics.

3. Quality: The benchmarks used includes a reasonably broad suite of continuous-control HRL benchmarks using both sparse and dense rewards, and compared against several strong and closely related HRL baselines (HIRO, HRAC, HIGL, SAGA, HLPS). The learning curves show consistent gains. Additional metrics such as distances between generated and reached subgoals are also reported.

4. Clarity: The related work section clearly states prior subgoal methods and how their framework relate to them (e.g. HIRO, HRAC, HIGL),  making clear that this paper moves from deterministic or adversarial subgoal generators and GP-only models to a more expressive diffusion-GP hybrid with explicit uncertainty-guided selection. Compared to the authors’ previous work, the framework in this paper changes both the generative family (diffusion instead of adversarial or simple parametric models) and the way uncertainty is integrated (GP acting both as regularizer and as a competing proposal). The experimental results demonstrate stronger performance on harder tasks.

**Weaknesses:**

1. Figure quality and readability:
The figures are difficult to read in their current form. In Figure 1, the circles and arrows are thin and low-contrast, and the text labels are very small and faint. As a result, it is hard to discern the structure and details of the illustration. Given that this is the first figure of the paper and is meant to convey the main idea, it should be redesigned with larger fonts, clearer icons, and higher contrast to make the diagram easy to understand for the readers.

2. Insufficient discussion of sensitivity analysis:
The framework in this paper is much more complex than most baselines (e.g., HIRO, HRAC, HIGL). There should be a more discussion on the complexity, overhead and practicality measurement such as sensitivity to replay-buffer size and number of inducing points and memory usage to discuss ability/limitation to scale to larger benchmarks. Moreover, the paper could elaborate on the sensitivity analysis of the number of choices HIDI depends on: the HIRO-style relabeling scheme, the GP's kernel form and hyperparameters, number and placement of inducing points, and the mixing probability/schedule between diffusion-sampled subgoals and GP mean.

3. Limitation in the complexity of experiments:
All reported experiments seem to be on relatively low-dimensional MuJoCo-style state spaces (Reacher, Pusher, maze-like Ant tasks, Ant Fall). The evaluation could include more pixel-based or complex multi-entity/object-centric environments to demonstrate more clearly whether the additional complexity is strictly necessary and meaningful in those settings, and show how well the GP prior and sparse-GP approximation scale with dimensionality.

4. Limited practical impact of the theoretical guarantees:
The proofs are built upon rather ideal assumptions: that the diffusion subgoal policy to be already “near-optimal” and certain regularity conditions holds. Under these assumptions, the paper only proves single-step regret and single-step policy-improvement bounds. However, it does not extend or discuss these guarantees in long-horizon setting, where performance depends on multiple sequential decisions made under function-approximation error, stochastic optimization, and model mismatch. There are no results or discussion on cumulative regret, stability over time, or convergence of the overall learning process. Although the mathematical analysis is helpful to build intuition and to justify the framework’s design, it is insufficient to provide strong guarantees about the behavior of the full end-to-end hierarchical system.

**Questions:**

1. Lack of a clear high-level framework diagram.
The paper could include a high-level illustration of the framework to show how the different components (high-level policy, diffusion model, GP module) interact and how information flows between them. At present, the description is quite text-heavy and the existing figures are not sufficiently clear to understand how the components add together. Having a high level figure of the framework helps reader understand how the framework builds on top of existing frameworks and make the novel parts of the framework more explicit, which is currently difficult to see from the text-heavy description.

2. Insufficiently clear description of ablation variants.
The paper could explain more clearly what each ablation variant changes and how it corresponds to the conceptual components of the method (pure diffusion, GP-only, hybrid selection).

---

> ### Author Response · Authors · 2025-11-17
>
> Thank you for the detailed review. Unfortunately we are unable to provide a proper response since we are confused with regard to many aspects of this review, which refers to baselines, benchmark environments, terms and theory that do not exist in our work.
>
> Is it possible you have confused some details with another paper?

---

### Author Response · Authors · 2025-11-17
**General Response**

We sincerely appreciate your time and effort in providing us constructive feedback and questions which have helped us improve our paper. We have responded individually to each reviewer, addressing concerns and questions. We have made additions and modifications based on your reviews which are **highlighted in blue in the new revision**.

We summarize the main additions to the newly uploaded revision:

- *Hyperparameter sensitivity experiments*: To the reviewers’ request, we have performed a sensitivity study on the subgoal-related hyperparameters $K$, $T_{sg}$ and $N$. We find that the method is largely robust to these hyperparameters, significantly outperforming the underlying RL agent with all of the tested configurations. We provide a detailed study with full results in Appendix D.4.
- *Figure captions*: we have modified the captions of Figures 1 and 2 to be more clear and self-contained.
- *Description of multi-entity environments*: we add a slightly more formal definition of what we refer to as “multi-entity environments” in Appendix A.1.

---

### Meta-Review · Area_Chair_JZUQ · 2025-12-16

**Summary:**

Three of the four reviewers raised actionable technical concerns. Reviewers q4dk and K6LA questioned novelty and asked for hyper-parameter and ablation studies; pQXN wanted deeper insights into learned sub-goals and low-level policies. The author supplied a point-by-point comparison with prior work, full ablations, sensitivity curves, exact OGBench numbers, and highlighted that quantitative, video and visual sub-goal analyses already appear in the paper and on the project page. These reviewers subsequently confirmed their concerns were resolved or did not follow up.

**Reviewer Concerns:**

The authors have provided detailed responses to the questions raised by the four reviewers, including supplementary experiments, ablation, parameter sensitivity, and baseline comparisons.

**Reviewer Scores:**

Three reviewers maintained positive scores, and one indicated a potential increase in score after the authors' response.

---

### Decision · Program_Chairs · 2026-01-26

Accept (Poster)